# Segmented multiblock polyolefin compatibilizers from non-living metathesis chain-shuffling

Abhishek Banerjee [1], Navin Kafle[1], Walter G. Romano[1], Harsh Pandya[1], Shin Horiuchi [2], Aarushi Srivastava[1], Fardin Khabaz [1], Mark D. Foster[1], Toshikazu Miyoshi [1] & James M. Eagan [1] ✉

The mechanical properties of mixed plastics can be enhanced by block copolymer compatibilizer additives. Here we report a series of semi-crystalline/semi-crystalline linear multiblock copolymers of LLDPE and PP, the two most abundant plastics, from a non-living shuffling polymerization. Containing up to 13 individual blocks of PE and PP randomly arranged, the multiblock connectivity, crystallinity, and morphology are characterized with a combination of NMR, DSC, X-ray scattering, microscopy, and theoretical models. We show these semi-crystalline products compatibilize and toughen blends of the two most abundant plastics with an enhancement of tensile modulus. Due to the highly segmented nature of the multiblock copolymers, the crystalline morphology exhibits minimal phase separation and imbibes the amorphous chains in commercial polymer interfaces into the semi-crystalline block copolymer phase by transmission electron microscopy. The work provides access to compatibilizer designs that implement a combination of tie-chain, trapped entanglement, and co-crystallization reinforcement mechanisms for toughening post-consumer blends.

Olefin polymerization is an energy and atom-efficient method for producing lightweight, strong, durable, and inexpensive macromolecules that are mechanically recyclable[1,2]. For these reasons, isotactic polypropylene (iPP, 21%, #5), low-density polyethylenes (LDPE and LLDPE, 20%, #4), and high-density polyethylene (HDPE, 16%, #2) are the most widely produced and utilized synthetic polymers[3]. They are also the principal components of post-consumer plastic waste (71%)[4]. Optical, infrared, and flotation separations are challenging for these polyolefins because of their similar plastic products, microstructures, and densities[5]. This is especially the case for iPP and LLDPE, wherein both microstructures contain alkyl branches and the two polymers are often intentionally mixed in virgin resins (e.g., impact-modified iPP). Whereas the alkyl branches in LLDPE serve to break up the crystallinity of the polymer, the stereospecificity of iPP enhances

crystallinity and is responsible for its desirable stiffness. The modulus, yield point, and ultimate tensile strength of iPP are its distinguishing properties that should not be compromised during the recycling process[6].

The compatibilization of mixed polyolefins is a pervasive strategy for improving the toughness of mixed secondary recycled plastics[7,8]. Polyolefin compatibilizers can be generalized as either reactive additives or non-reactive reinforcing agents[9,10]. Non-reactive block copolymers are particularly useful as they preserve the reprocessability of the materials without necessarily increasing the complexity of the secondary mixture with exogenous species, such as filler particles, ions, plasticizers, or polar moieties[11].

Non-reactive copolymers reinforce PP/PE interfaces by increasing polymer chain entanglements across the phase boundary, co-

[1]The School of Polymer Science and Polymer Engineering, The University of Akron, Akron, OH, USA. [2]Research Laboratory for Adhesion and Interfacial Phenomena, Nanomaterials Research Institute, National Institute of Advanced Industrial Science and Technology, Tsukuba, Ibaraki, Japan. ✉e-mail: eagan@uakron.edu

crystallizing with their respective phases, and/or acting as tie-chains between crystalline domains (Fig. 1)[11]. The operating mechanism of reinforcement, the compatibilizer efficacy, and the resulting mechanical properties of the blend are dependent on the compatibilizer's polymer architecture, microstructure, and crystallinity. Linear multiblock compatibilizers have proven more efficient under a wider range of processing methods than diblock copolymers[12–16], due to their ability to form trapped entanglements between the two polyolefin phases, although the relative importance of each mechanism (i.e., trapped entanglements, tie-chains, and co-crystallization) is not entirely understood.

Synthetically, linear multiblock polyolefin materials are exclusively produced by living polymerization and/or contain rubbery blocks[13,16,17]. This results in large amounts of initiators required or a loss of stiffness in the plastic. To address these challenges, coordinative chain-transfer polymerization or graft-through polymerization was developed for the synthesis of semi-crystalline/semi-crystalline diblock and graft copolymers[18–24]. Conceptually, the diblocks and graft copolymers produced from these methods suffer from fewer trapped entanglements at the interface, since the compatibilizers are no longer linear architectures, and thus the chain ends are susceptible to pull-out under applied stress[21,25–27].

Another identified limitation in existing compatibilizer additives is the absence of iPP-LLDPE block copolymers, the two most abundant plastic microstructures. We recently reported the synthesis of iPP-*b*-LLDPE diblock copolymers using living polymerization through a sequential terpolymerization of propylene, followed by ethylene and octene[28]; but this method is limited to diblocks and cannot access multiblocks. The synthetic limitation is due to the addition of the alpha olefin comonomer (e.g., 1-octene), which is incorporated at low conversions and therefore is incorporated into subsequent iPP blocks, disrupting the crystallinity.

Motivated by the need for a non-living and versatile multiblock semi-crystalline polyolefin synthetic method, we report a method for producing previously inaccessible iPP-*b*-LLDPE linear multiblocks wherein the overall molecular weights, individual block lengths, and crystallinity can be controlled. The approach utilizes independently synthesizing unsaturated iPP (uPP) and unsaturated LLDPE (uPE), which are cleaved into telechelics and subsequently chain-extended using a non-living metathesis step-growth polymerization reaction to yield randomly segmented block copolymers. The overall process of shuffling semi-crystalline PE and PP yields multiblock polymers with homogeneous morphologies, which act as effective toughening agents in brittle iPP/LLDPE blends while simultaneously enhancing the tensile modulus through interfacial crystallization and trapped entanglements. The findings also reveal design parameters and compatibilization mechanisms in the two most abundant plastics produced.

## Results

The synthetic strategy begins with the synthesis of unsaturated iPP and LLDPE chains, which are cleaved into telechelic polyolefins using metathesis, and then randomly reconnected through acyclic diene metathesis (ADMET) step-growth polymerization. The general approach was applied by Fredrickson and Bazan for the synthesis of LLDPE-*b*-poly(ethylene-ran-ethylethylene) multiblock polymers, wherein the blocks are hydrogenated polybutadienes (pBD) with varying degrees of 1,4 and 1,2 insertions[29,30]. Herein, we instead employ semi-crystalline telechelic iPP chains to produce compatibilizers and semi-crystalline interfacial tie-chains (Fig. 2). Notably, olefin metathesis avoids exogenous polar moieties, which have shown detrimental effects on compatibilization efficacy[31], and maintains the desirable hydrocarbon composition of commercial polyolefins.

The synthesis of telechelic LLDPE (**9**), was carried out using partial hydrogenation of commercial high-cis 1,4-pBD (**3**) using Wilkinson's catalyst (**4**) and ethenolysis using Ru catalyst **1**[32]. The hydrogenation of commercial pBD (98% cis 1,4-enchainment, Supplementary Fig. S1) at 380 kPa of $H_2$ in toluene at 75 °C produced unsaturated PE (uPE, **5**) with 2.7 mol% of the ethylene repeat units containing internal 1,2 disubstituted alkene moieties by [1]H NMR (Supplementary Fig. S3). Metathesis cleavage using catalyst **1** (0.4 mol% to uPE alkenes) in PhMe at 90 °C afforded 54% conversion of the internal 1,2-disubstituted alkenes to telechelic chain ends. This could be improved to full conversion through sequential catalyst additions (i.e., 3 additions of 0.4 mol% Ru). However, with the aim of reducing catalyst consumption for multiblock copolymer production, we carried the 0.4 mol% materials forward, despite the residual internal unsaturation. The [13]C NMRs of pBD, uPE, and the telechelic LLDPE are consistent with the structural assignments in Fig. 2 and further revealed that >98% of the chains are telechelic with less than 2 mol% of chain ends containing unreactive $CH_3$ moieties (Supplementary Fig. S6). The SEC analysis (Supplementary Fig. S13) revealed uPE ($M_n$ = 211,900 g/mol) was cleaved into fragments of $M_n$ = 4600 g/mol ($M_w$ = 11,500 g/mol). The [1]H NMR molecular weight was 4760 g/mol, in agreement with the SEC and supports the high fidelity of telechelic end-groups. The melting points ($T_m$) of the uPE and telechelic LLDPE were 124 and 122 °C; the crystallization temperatures ($T_c$) were 102 and 112 °C, respectively.

Telechelic iPP, however, is both more challenging to synthesize and is less studied[33–37]. To obtain sterically accessible allyl end-groups, we selected the copolymerization of propene and 1,3-butadiene to produce unsaturated polypropylene (uPP). Current approaches to uPP suffer from low productivity (TON = 500–3800 mol PP/mol cat)[33] due to the metal catalyst π-allyl resting state or low crystallinity ($T_m$ < 110 °C)[37]. We achieved stereoselective ($m^4$ = 92.5%, $T_m$ = 131 °C, 82% 1,4-selective), productive (TON = 8450) uPP polymerization by employing catalyst **6**/PMAO-IP in PhMe at 0 °C under a constant feed of propylene (275 kPa). Notable optimizations included maintaining a constant concentration of propene to avoid accumulation of the butadiene-derived Zr π-allyl terminating species, and PMAO-IP cocatalyst, which disfavored Zr/Al trans-metalation. The resulting uPP was cleaved to telechelic iPP using analogous conditions to the PE materials (0.4 mol% Ru to alkene) to afford full conversion of the internal alkenes to allyl and homo-allyl moieties (**8**, Fig. 2) by [1]H and [13]C NMR (Supplementary Figs. S9, S10). Interestingly, upon precipitation of the telechelic iPP, we observed mass loss in the form of MeOH-

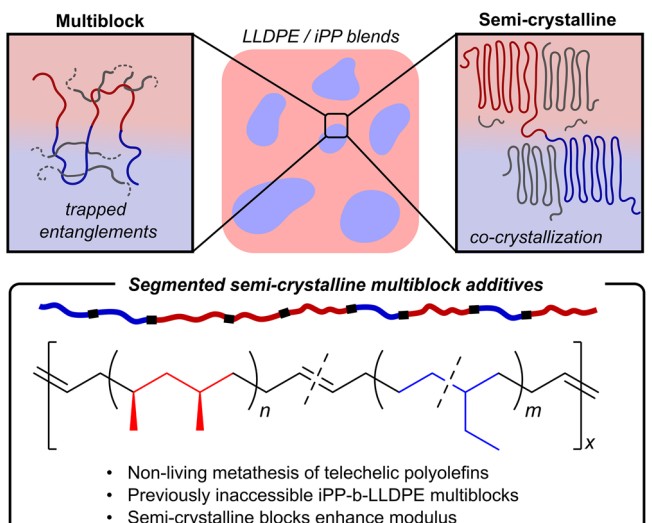

**Multiblock**       *LLDPE / iPP blends*       **Semi-crystalline**

*trapped entanglements*       *co-crystallization*

***Segmented semi-crystalline multiblock additives***

- Non-living metathesis of telechelic polyolefins
- Previously inaccessible iPP-b-LLDPE multiblocks
- Semi-crystalline blocks enhance modulus

**Fig. 1 | The compatibilization of mixed polyolefin plastics.** Non-reactive block copolymer additives can be designed with a multiblock structure possessing trapped entanglements and semi-crystalline blocks that co-crystallize with the plastics. In this work, the non-living synthesis of linear multiblock copolymers are synthesized and studied as a compatibilizers.

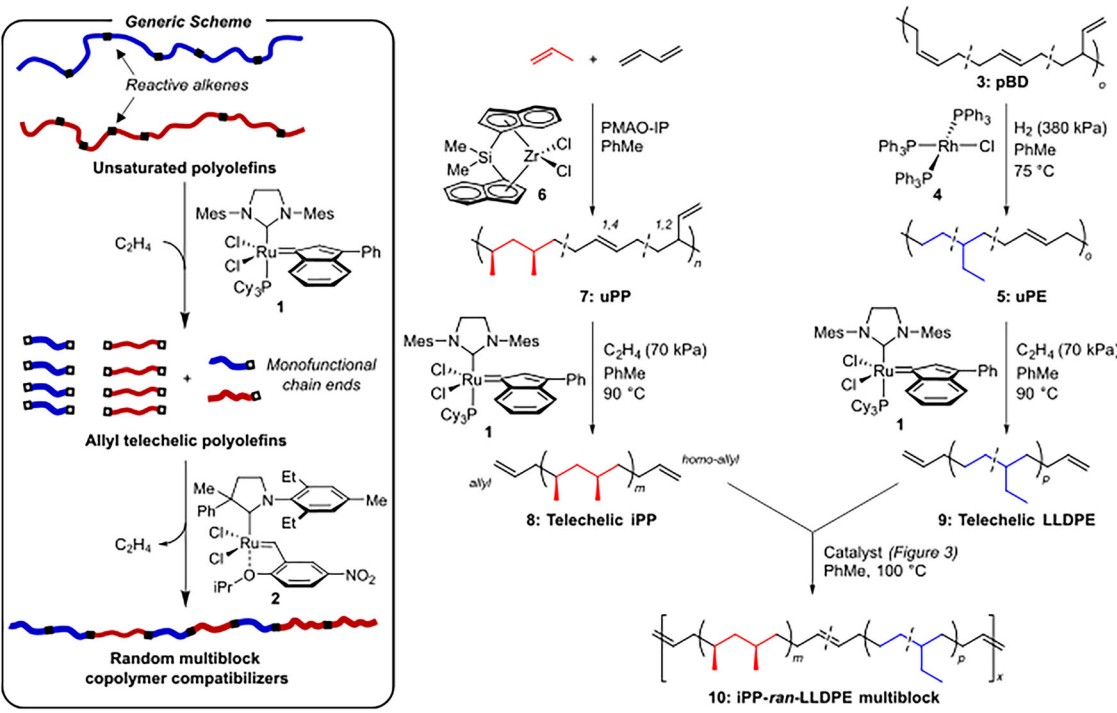

**Fig. 2 | The chain-shuffling synthetic sequence.** The generic scheme and specific procedures for producing unsaturated semi-crystalline polyolefins, metathesis cleavage of these polymers to telechelics, and the polymerization into segmented multiblock polymers.

soluble oligomers (c.a., 10 wt%), which can be attributed to the localization of butadiene linkages in the uPP chains. The SEC analysis (Supplementary Figs. S15, S16) showed the ethenolysis reaction reduced the molar mass of uPP ($M_n$ = 31,300 g/mol, $M_w$ = 63,300 g/mol) to 6400 g/mol ($M_w$ = 10,300 g/mol); and the high temperature solution [1]H NMR analysis was consistent with an average molecular weight of 5200 g/mol based on telechelic allyl end-groups, indicating difunctionality of the polymer chains. Undesired unfunctionalized chain-ends were below the solution [13]C NMR limit of detection, while the target unsaturated ends could be clearly identified as the homoallyl and allyl chain-end moieties at δ = 113.6, 139.1 and 115.0, 137.2 ppm, respectively (Supplementary Fig. S10).

With telechelic iPP (**8**) and LLDPE (**9**) in hand, the metathesis shuffling polymerization was next developed with the aim of increasing molecular weights and cross-reactivity between phases (Fig. 3). The reactions were conducted by dissolving the telechelic iPP and LLDPE precursors in a 1:1 weight ratio in toluene at 100 °C under nitrogen, injecting the metathesis catalyst, and evaporating the solvent under a positive flow of nitrogen over 30 min. This approach maximized intermixing of the PE and PP phases through solvation and the concentration of the end-groups for promoting the polymerization in what is effectively a bulk solid-state polymerization.

Using the above procedure, we studied a series of metathesis catalysts (Fig. 3b). WCl$_6$/SnMe$_4$, which had previously been studied by Ishihara et al., was both excessive in hazardous metal quantities (100 equiv. to alkene) and ineffective at achieving conversion of the telechelic macromolecules[33]. Similar to Fredrickson and Bazan[29,30], ruthenium catalyst **1** achieved conversion at 1.6 mol% loading relative to alkene, however, molecular weights were relatively low for the PE/PP system ($M_w$ = 68,500 g/mol), corresponding to approximately 6 blocks randomly linked together. We assessed the number of blocks using the weight-average degree of polymerization ($DP_w$) from the ratio of the molecular weight ($M_w$) of the multiblock and the average $M_w$ of the telechelic precursors (Fig. 3d). We additionally investigated phosphine-free ruthenium catalysts **11**, **12**, and **14**, which produced low $M_w$ products (25, 42, and 52 kDa, respectively). Cyclic alkyl amino

carbene ruthenium catalysts **13** and **2**, on the other hand, yielded products with molecular weights of 43 and 76 kDa, respectively, at loadings of 1.6 mol% Ru to alkene. Fogg and coworkers originally developed catalyst **2** and demonstrated increased reactivity in ethenolysis reactions and at elevated temperatures due to the improved stability of the intermediate ruthenium methylidene[38]. Note, LLDPE and iPP are semi-crystalline polyolefins that mandate chemistry capable of high productivity at temperatures >80 °C to avoid precipitation, temperatures which most Ru catalysts are prone to deactivation. When the catalyst loading of **2** was increased from 1.6 mol% relative to alkene chain ends to 2.2 mol% we found the $DP_w$ for the multiblock increased to 13 blocks (Fig. 3d). The SEC analysis (Fig. 3c) revealed that the telechelic polymers were consumed into higher molecular weight polyolefins ($M_w$ = 143,000 g/mol). It should not be overlooked that the number average molar mass of this multiblock ($M_n$ = 39,100 g/mol) suggests a number average degree of polymerization ($DP_n$) of 5 blocks. This is a critical threshold for compatibilization mechanisms (i.e., ≥4) wherein both PE and iPP phases have internal trapped entanglements. However, since compatibilizers are employed as wt% additives, we assert the $M_w$ and $DP_w$ are more descriptive measures. When the ruthenium loading was further increased to 3.7 mol%, the multiblock $M_w$ decreased due to an increase in monofunctional benzylidene chain ends introduced by the catalyst. However, if too little catalyst is used (i.e., 0.6 mol% per alkene moiety), the $M_w$ severely decreases to only 19,300 g/mol. We expect this is due to the catalyst losing activity because of trace contaminations and deactivation, most likely because of residual aluminum salts from the MAO co-catalyst in the iPP synthesis. It is important to note the apparent molecular weights were measured by size-exclusion relative to monodisperse polystyrene standards adjusted by iPP's Mark–Houwink parameters.

The degree of unsaturation in the multiblock was determined by [1]H NMR to be 1.1 mol% of the overall repeat units (Supplementary Fig. S11), consistent with the degree of unsaturation in the telechelic PE and PP precursors (1.2 and 1.6 mol%, respectively) after metathesis. The ADMET chain-shuffling reaction is furthermore dependent on the uPP microstructural 1,4-selectivity and end-group fidelity. Currently, our

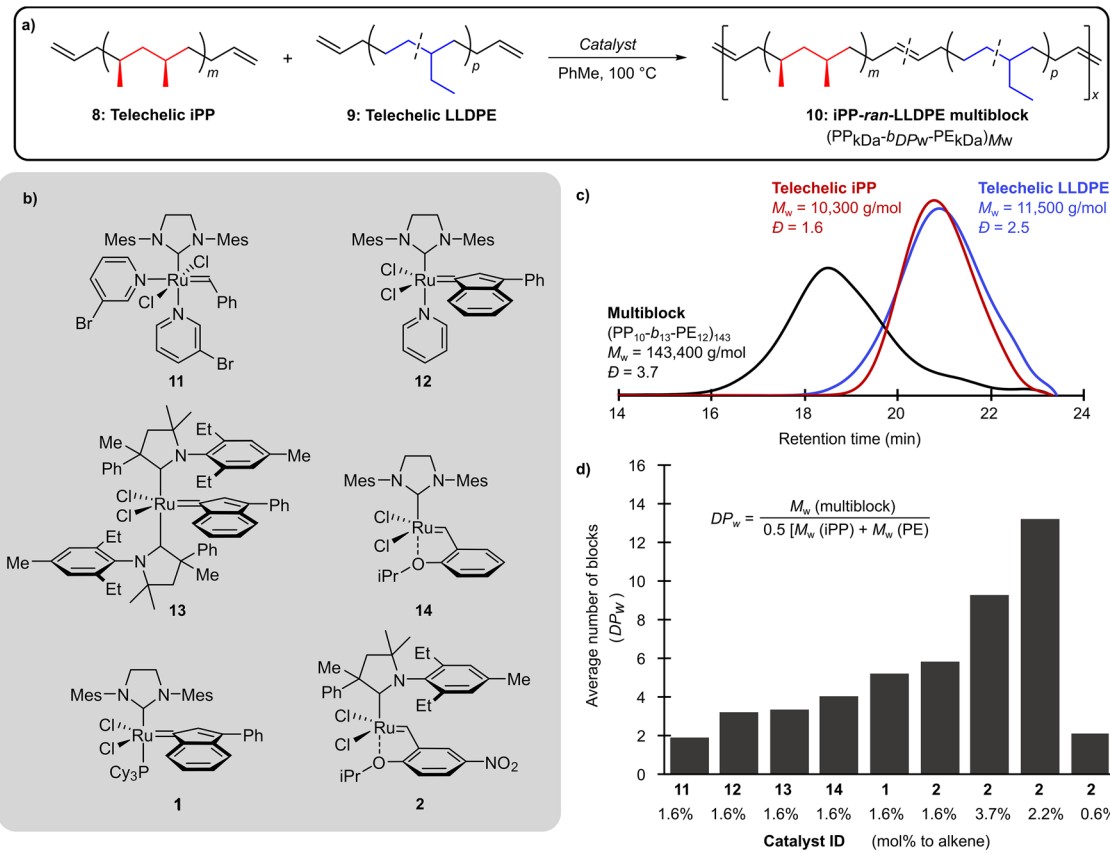

**Fig. 3 | Optimization of metathesis catalyst.** The chain-shuffling reaction to combine (**a**) LLDPE and iPP telechelics into multiblock polyolefins and the effects of (**b**) metathesis catalyst on the high-temperature size-exclusion chromatography (SEC) (**c**) and corresponding degrees of polymerization (**d**) for each individual experiment.

investigations have no evidence in support of, or excluding, the formation of cyclic oligomers from intramolecular cyclization or hyperbranched structures from the <10% of unsaturated units in the combined telechelics that are 1,2-vinyl microstructures. Furthermore, the step-growth polymerization of telechelic chains will be terminated by monofunctional chains acting as capping blocks, according to the Carothers equation (Fig. S26)[39]. Although unfunctionalized chain-ends were below the limit of spectroscopic detection, it may be supposed that a 6-fold reduction in molecular weight during telechelic iPP synthesis affords 17% monofunctional chains. During the shuffling reaction, these capping blocks will limit the maximum $DP$ at 13, which is in close agreement with the experimental SEC measurements. We therefore expect the $DP_w$ is limited by the iPP telechelic fidelity under these conditions.

The SEC data, however, provides no insights into the randomness of the blocks or even if the blocks are indeed covalently linked, rather than existing as a mixture of unsaturated homopolymers (i.e., each independently reverting back to uPP or uPE within their respective phase). The materials were characterized using high-temperature solution NMR to investigate the alkene resonances in telechelic precursors, homopolymer products, and multiblock products (Supplementary Figs. S12–S14). The ¹H NMRs of telechelic LLDPE ($M_w$ = 11,500 g/mol), telechelic iPP ($M_w$ = 10,300 g/mol), and the 1:1 wt. Ratio multiblock ($M_w$ = 143,400 g/mol) shows full conversion of the telechelic terminal alkenes into internal 1,2-disubstituted alkenes. Despite the broad ppm range of ¹³C NMR, only limited signals could be identified with a robust signal-to-noise ratio, even after 48 h acquisition times with 187.5 MHz solution ¹³C NMR (750 MHz, ¹H). One reason for the reduced intensity of the signals is the dilution of iPP ¹³C resonance to regio- and stereoisomeric linkages of the allyl, homo-allyl, and hetero-linkage resonances (Supplementary Fig. S14). This explanation

is supported by a control study of deliberately synthesizing [PP-*b*-PP]ₓ from the telechelic iPP, which exhibits new olefinic resonances compared to the initial uPP due to these isomeric linkages. The differences between the [PP-*b*-PP]ₓ homopolymer and the [PP-*b*-PE]ₓ multiblock ¹³C NMRs (Supplementary Fig. S14) support the conclusion that homopolymerization is not the dominant process.

Figure 4a presents a predictive Monte-Carlo model for relating the % yield of various multiblock architectures for varying degrees of polymerization ($DP_w$), assuming equal reactivity of the telechelic chains and step-growth polymerization mechanisms. For example, at $DP_w$ = 2, the macromonomers have combined to form diblocks in a statistical 50% yield along with a 50% yield of uPE and uPP homopolymers. According to the step-growth polymerization mechanisms of ADMET, these linear diblocks serve as monomers for reaching a $DP_w$ = 4, and the degree of blockiness likewise increases, although only a 12% yield of tetrablock is expected. The degree of blockiness is assessed by quantifying the number of alternating macromonomer segments (i.e., PE-*b*-PP linkages) and simplifies homo-linkages as larger single blocks. Therefore, as the $DP_w$ increases, not only do the overall molecular weights increase, but the materials become increasingly segmented and blocky. At the experimentally obtained values of $DP_w$ = 13, this ideal statistical model predicts predominantly octa-blocks to be present with <2% existing as homopolymer, diblock, or triblocks incapable of forming trapped entanglements in both phases.

In addition to the scalable non-living nature of the chain-shuffling method, there is experimental convenience and versatility in the method for independently controlling the minimal block length and overall blockiness of the products. Table 1 demonstrates a series of (PE_kDa-*b*_DPw-PP_kDa)_MW multiblocks wherein the kDa subscripts correspond to the molecular weights of the PE and PP blocks, the $DP_w$ block subscript corresponds to the average number of blocks, and the

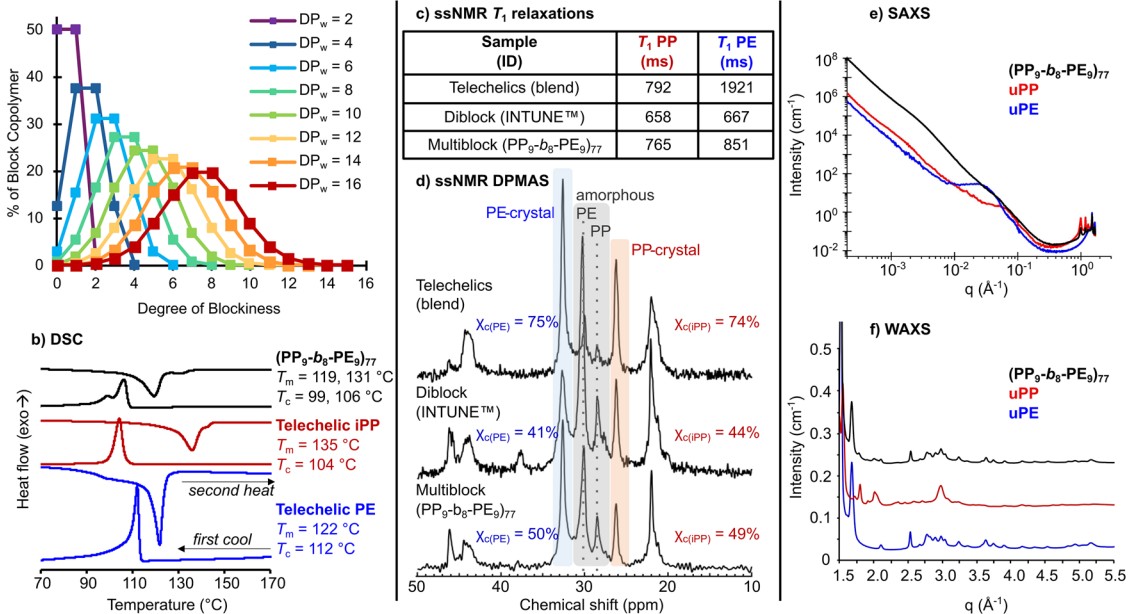

**Fig. 4 | The characterization of multiblock structure and crystallinity compared to precursors and diblock copolymer. a** The theoretical percentage of linear segmented block copolymers as a function of their degree of polymerization ($DP_w$) as evaluated by the number of alternating macromonomer segments (i.e., degree of blockiness). **b** The melting temperatures ($T_m$), crystallization temperatures ($T_c$) of the multiblock copolymer and its telechelic precursors as measured by DSC. **c** Solid-state NMR $^1$H relaxation times through highly resolved $^{13}$C signals of blends, diblocks, and multiblock copolymers showing coupling or disparate relaxations as a result of nanoscale mixing, and **d** the crystallinity of each component as measured by $^{13}$C direct polarization magic-angle spinning (DPMAS). **e** The small-angle X-ray scattering and **f** wide-angle X-ray scattering of the multiblocks and their high molecular weight unsaturated precursors. WAXS data for uPP and multiblock samples are vertically shifted by 0.1 cm$^{-1}$ and 0.2 cm$^{-1}$, respectively, for visual clarity.

## Table 1 | LLDPE/iPP block copolymer compatibilizers from non-living metathesis shuffling of unsaturated semi-crystalline precursors

| Entry (#) | Polymer (PE$_{kDa}$-b$_{DPw}$-PP$_{kDa}$)$_{MW}$ | $M_w$ (g/mol)$^a$ | Đ ($M_w/M_n$)$^a$ | $T_m$ (°C)$^b$ | $T_c$ (°C)$^b$ |
|---|---|---|---|---|---|
| 1 | (PE$_{12}$-b$_{13}$-iPP$_{10}$)$_{143}$ | 143,400 | 3.7 | 119, 131 | 99, 106 |
| 2 | (PE$_{12}$-b$_9$-iPP$_{10}$)$_{99}$ | 98,600 | 2.8 | 119, 131 | 100, 107 |
| 3 | (PE$_5$-b$_4$-iPP$_{10}$)$_{31}$ | 31,300 | 3.1 | 115, 129 | 96, 103 |
| 4 | (PE$_9$-b$_8$-iPP$_9$)$_{77}$ | 76,600 | 4.2 | 115, 130 | 94, 101 |

$^a$Weight average molecular weight and distributions determined from high temperature size-exclusion chromatography (SEC) in 1,3,5-trichlorobenzene at 140 °C relative to monodisperse PS standards adjusted by Mark–Houwink factors to iPP.
$^b$Melting and crystallization temperatures measured from differential scanning calorimetry second heating cycle and first cooling cycle at 10 °C, respectively.

parenthetical $M_W$ subscript is the overall molecular weight of the multiblock measured by high-temperature SEC. With the described synthetic approach, the minimum block lengths were controlled by the degree of unsaturation in the polyolefin precursors and the overall molecular weight by the metathesis catalyst loading. All the chain-shuffling products have an equal weight ratio of iPP and LLDPE, regardless of their individual block lengths, and exhibit two $T_m$s and $T_c$s corresponding to the iPP and LLDPE crystallites.

Next, in order to experimentally support our expectation of segmented multiblocks having formed, we opted to investigate the phase structure of the multiblock polymer using solid-state NMR (ssNMR). Relative to solution NMR, ssNMR permits higher concentrations of analyte and can probe the spatial proximity between active $^1$H nuclei. Specifically, ssNMR $^1$H relaxation experiments ascertained a degree of mixing and magnetic resonance between the blocks at 20–30 nm length-scales. We first investigated the $^1$H $T_1$ relaxation of the crystalline domains through the highly resolved $^{13}$C signal in each the iPP and LLDPE phase for a blend of telechelics (1:1 wt. ratio), the chain-shuffled

multiblock (PE$_{12}$-b$_9$-iPP$_{10}$)$_{99}$, and a commercial PE-b-iPP diblock (Dow's INTUNE™). Figure 4c shows how when the semi-crystalline polymers were macrophase separated as a blend of homopolymers, the $T_1$ values exhibit no consistency with each other (PP = 792 ms, PE = 1921 ms). This means the two components are physically separated from each other. However, in the commercial diblock copolymer, we observed $T_1$ relaxations that were closely coupled (PP = 658 ms, PE = 667 ms) as a result of the nanoscale mixing that promotes $^1$H–$^1$H spin diffusion between PE and PP crystalline regions. The multiblock polymer obtained from the chain-shuffling reaction also showed close relaxations (PP = 765 ms, PE = 851 ms), which is attributed to the covalent hetero-linkages and nanoscale intermixing of crystalline lamellae.

As previously mentioned, the crystallinity of block copolymer compatibilizers is critical to the simultaneous optimization of both toughness and stiffness. Using ssNMR DPMAS techniques, the amorphous and crystalline regions of both the PP (28.5 and 26.0 ppm, respectively) and PE (30.0 and 32.5 ppm, respectively) can be resolved (Fig. 4d). This enables the integration and determination of the degree of crystallinity ($X_c$). The crystallinity of the PP and PE telechelics ($X_c$ = 74% and 75%, respectively) was considerably decreased by the shuffling reaction (49% and 50%, respectively), which is due to the hetero-linkages between blocks being forced to reside in an amorphous phase at the PE/PP interfaces. Interestingly, despite the decrease in crystallinity due to blockiness and the use of LLDPE blocks, these multiblock materials are still more crystalline than commercial diblocks of PP and PE (44% and 41%, respectively). The higher crystallinity in the chain-shuffled multiblock results from the defined block compositions and the absence of tapering between PP and PE blocks, a rubbery domain which is formed in some sequential addition chain-growth polymerizations[40]. We further confirmed the crystallinity measurements using proton broadline spectra (Supplementary Fig. S17) and wide-angle X-ray diffraction (Supplementary Fig. S18), which provided estimates of overall $X_c$ for the multiblock of 55% and 49% for proton broadline spectra and WAXS, respectively.

Small-angle X-ray scattering (SAXS, Fig. 4e) and wide-angle X-ray scattering (WAXS, Fig. 4f) were used to characterize the intermixed crystalline morphology of the multiblock and its precursor homopolymers. Both the multiblock and telechelic materials exhibit WAXS patterns characteristic of the iPP monoclinic and PE orthorhombic phases. This indicates that despite the segmented nature of the multiblock, both the iPP and LLDPE blocks crystallize independently in their crystalline phases despite the relatively short block sizes of ($M_w$ ~10,000 g/mol). This supports the observation of two $T_m$s during the DSC heating and cooling cycles, respectively. The SAXS profile, however, is notably different in the multiblock sample relative to the two homopolymers. The iPP and LLDPE homopolymer $q_{max}$ values were 0.035 and 0.040 $\text{Å}^{-1}$, corresponding to interlamellar spacings ($d_1$) of 18.0 and 15.7 nm, respectively, according to the Bragg equation $d_1 = 2\pi q_{max}^{-1}$. The multiblock has much broader $q_{max}$ values, such that they are almost indistinguishable even on a log–log scale, indicating the interlamellar distances are much smaller than in the homopolymers. This is consistent with other multiblocks in the literature prepared from precise living polymerization[12]. Unique to the shuffled multiblocks, the scattering invariant increases two orders of magnitude for the shuffled segmented multiblock as a result of the greater number of domains and interfaces between the iPP and LLDPE lamellae.

The high stress at yield and strain at break for PE and ductile iPPs are a result of high crystallinity and tie-chains within the load-bearing polymer network, distributing applied stresses throughout the material[41]. When PE and PP are mixed, the non-continuous interface interrupts the physical network and results in brittle materials. Notable compatibilizers previously studied include triblock poly(ethylene)-*b*-poly(ethylene-co-ethylethylene) polymers, which demonstrated that a large, extensively entangled amorphous mid-block was critical for blend ductility[16]. Living semi-crystalline tetrablock and hexablock (HDPE-*b*-iPP)$_x$ materials contain long block segments >10-fold $M_e$ and are effective compatibilizers even at 0.2–0.5 wt.% loadings[12,13]. Graft PE/PP copolymers have also shown compatibilization efficacy through a proposed tie-chain mechanism[20,21,42]. We recognized that chain-shuffled segmented multiblocks were a unique opportunity to invoke the combined effects of trapped entanglements, tie-chain, and co-crystallization.

The stress-strain properties of iPP, LLDPE, and their blends were studied with and without the chain-shuffled multiblock additives (Fig. 5). The blends were mixed with a counter-rotating batch micro compounder at 185 °C for 7 min, and then all tensile specimens were prepared by compression molding at 180 °C for 3 min, followed by rapid cooling in air. The commercial polyolefins, which were produced from heterogeneous Ziegler−Natta catalysts, showed ductile behavior when strained at 100% $\text{min}^{-1}$ to reach a maximum strain at break of 619 ± 25 and 569 ± 73% for LLDPE and iPP, respectively. The Young's moduli were 332 ± 14 MPa for LLDPE and 823 ± 82 MPa for iPP. The similar global production volumes of iPP and LLDPE (21% and 20%, respectively) prompted us to investigate a 50:50 wt. ratio blend, a mixture which is also the least ductile[43]. The resulting compression-molded blends were brittle with low strains at break (124 ± 54%) and possessed low modulus (541 ± 51 MPa). When a high degree of polymerization ($DP_w = 13$) multiblock (PE$_{12}$-*b*$_{13}$-PP$_{10}$)$_{143}$ was added in 5 wt.%, the elongations increased to >700% (Fig. 5a). The modulus of the blends also considerably increased up to 685 ± 24 MPa at 5 wt.% of the crystalline additive. Less efficient toughening was observed with a moderate sized multiblock ($DP_w = 9$); elongation at break was 724% and the modulus was 687 ± 68 MPa at 5 wt% additive. When the low molecular weight multiblock (PE$_5$-*b*$_4$-PP$_{10}$)$_{31}$ was employed at 5 wt%, however, we observed brittle blends that only elongated to 124 ± 54%. This results from saturating the interface with the low molecular block copolymer, which cannot entangle with the bulk polyolefin network as a tie-chain. This is exacerbated by the shorter PE block ($M_w = 5$ kDa),

which is only ~$5M_e$. However, this finding importantly emphasizes that low molecular weight compatibilizers−even if able to form trapped entanglements and co-crystallites−are less effective at integrating into the bulk polyolefin load-bearing network, and therefore, tie-chain mechanisms are critical for compatibilization. This is consistent with the thread-the-needle mechanism and suggests co-crystallization may not contribute to extensibility. The optimal high molecular weight multiblock (PE$_{12}$-*b*$_{13}$-PP$_{10}$)$_{143}$ could be reduced to 1 wt% loading without sacrificing extensibility (Fig. 5a), but lost ductility at 0.5 wt% and 0.2 wt % (Supplementary Fig. S25).

The morphologies of the blend were studied by scanning electron microscopy (SEM) after brittle fracture in liquid nitrogen (Supplementary Fig. S27). The 50:50 blend containing no additive had a distribution of droplets with an average diameter of 2.06 ± 1.19 μm. There was evidence of coalescence of the droplets during mixing into a co-continuous morphology. Upon the addition of 1 and 5 wt% of multiblock (PE$_{12}$-*b*$_{13}$-PP$_{10}$)$_{143}$, the distinguishable droplets exhibited a slight reduction in average diameters to 1.92 ± 0.77 μm and 1.67 ± 0.73 μm, respectively (Supplementary Fig. S27). This droplet reduction is much less than in HDPE/iPP blends[13] and suggests that droplet size reduction is not solely responsible for toughening these blends.

Closer examination of the stress at yield provides further insights into the mechanism and distinguishing utility of semi-crystalline/semi-crystalline multiblock compatibilizers. A 50:50 blend of LLDPE (10 ± 0.5 MPa) and iPP (32 ± 2.4 MPa) has a theoretical yield stress of 21 MPa, which the measured stress at yield of 17 ± 1.7 MPa deviated negatively from. This is consistent with previous LLDPE/iPP blends previously studied, where a negative deviation from mixing rules is observed in yield stress[44,45]. This effect was counteracted upon the addition of the semi-crystalline multiblock copolymer, and the stress at yield increases up to 21 ± 0.6 MPa. We propose that the observation that Young's modulus and yield stress increase upon the addition of results from the interfacial crystallization in these semi-crystalline/semi-crystalline additives and supports their usefulness in the mechanical recycling of rigid iPP.

In order to establish if the PE and PP blocks co-crystallize with the commercial homopolymers, the (PP$_{10}$-*b*$_{13}$-PE$_{12}$)$_{143}$ multiblock copolymer was solvent blended with Ziegler−Natta LLDPE or iPP in a 1:1 ratio and studied by DSC. The commercial polymers exhibited characteristic melting points (LLDPE $T_m$ = 107, 125 °C; iPP $T_m$ = 168 °C) and crystallization temperatures (LLDPE $T_c$ = 102 °C; iPP $T_c$ = 117 °C), and the multiblock compatibilizer exhibited $T_m$s and $T_c$s for both phases (LLDPE-block $T_m$ = 119 °C, $T_c$ = 106 °C; iPP-block $T_m$ = 131 °C, $T_c$ = 99 °C) −transitions which were assigned based on their telechelic precursors (Fig. 4b). Figure 5b shows the second heating and first cooling cycles for these materials and their blends. When the multiblock (PP$_{10}$-*b*$_{13}$-PE$_{12}$)$_{143}$ was blended with the commercial LLDPE, the $T_c$ and $T_m$ of the commercial materials shifted towards the higher values of the multiblock (107 and 121 °C, respectively). This indicates the multiblock induces the PE crystallization process in the commercial LLDPE; meanwhile, the iPP phase is unaltered and crystallizes separately. Conversely, when the multiblock was blended with commercial iPP, which has higher tacticity, the multiblock iPP $T_m$ of 131 °C and $T_c$ of 99 °C both disappeared. Instead, two melting transitions were observed at 155 and 162 °C, and a single $T_c$ of 113 °C was detected, indicative of the co-crystallized morphologies, in this case dominated by the Ziegler−Natta iPP material. The PE blocks were unaltered in the PP blend. These DSC results of a model interface support the conclusion that co-crystallization between the multiblock compatibilizer occurs in the compatibilized blends.

An alternative explanation of the observed properties could be through a reactive mechanism involving the unsaturated alkenes in the multiblocks. Polyunsaturated polymers can undergo oxidative chain-scission or crosslinking, which would obscure our mechanistic rationale and limit the repeated use of the additives in recycling. To address

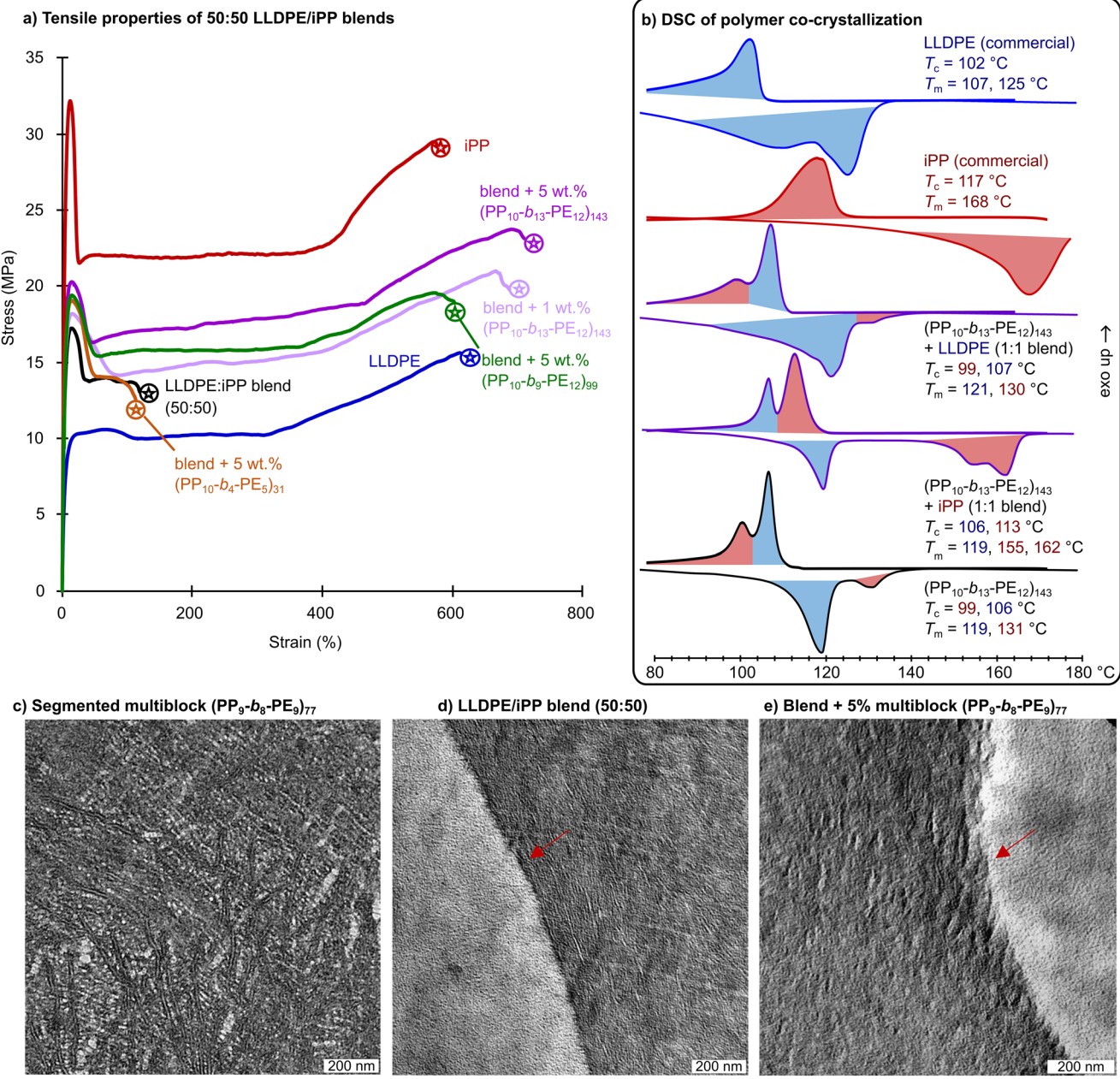

**Fig. 5 | The mechanical characterization of compatibilized blends and their semicrystalline morphologies. a** Uniaxial tensile tests at 100% min⁻¹of commercial Ziegler–Natta LLDPE, iPP, blends (50:50), and with the addition of multiblock compatibilizers. **b** DSC of the first cooling and second heating cycles (10 °C min⁻¹) for LLDPE, iPP, multiblock ($PE_{10}$-$b_{13}$-$PE_{12}$)$_{143}$ and their 1:1 blends demonstrating the co-crystallization of the multiblock phases with their respective commercial polyolefins. **c** Transmission electron microscopy of (**a**) pure multiblock copolymer ($PE_9$-$b_8$-$iPP_9$)$_{77}$, (**d**) uncompatibilized blend of LLDPE/iPP (1:1 wt. ratio), and (**e**) the same blend with 5 wt.% of added ($PE_9$-$b_8$-$iPP_9$)$_{77}$ multiblock copolymer. Red arrows added to draw attention to the interfaces containing (**d**) amorphous band of accumulated amorphous materials versus the (**e**) diffuse interface induced by semi-crystalline multiblock additive.

this, sol/gel measurements, stability studies, and post-polymerization modification of the alkenes were carried out to better understand the impact unsaturation has on the system. First, the high molecular weight multiblock ($PP_{10}$-$b_{13}$-$PE_{12}$)$_{143}$ was processed through several conditions aimed at simulating aging. The multiblock was subjected to the same melt-blending procedure used in preparing the PE-PP blends (185 °C, micro-compounder, 7 min, N₂), heating in xylenes (130 °C, 96 h, air), and prolonged ambient storage in air (20 months, air). Soxhlet extraction with toluene resulted in near complete solubility (> 95% Supplementary Fig. S22) and minimal changes in molar mass (< 6% change in $M_w$) for all conditions, indicating that both crosslinking and chain-scission are suppressed even without added stabilizing agents. Additionally, the stoichiometric hydrogenation of the multi-block with pTsNHNH₂ afforded a saturated multiblock copolymer which exhibited the same toughening properties at 1 wt.% loading (Supplementary Fig. 24). These results indicate the crosslinking and degradation due to the residual unsaturation (ca. 1.1 mol%) are negli-gible, likely due to the low concentration in the multiblocks and orders of magnitude lower concentrations within the blends. For these rea-sons, the blends containing 1 wt.% unsaturated ($PE_{12}$-$b_{13}$-$iPP_{10}$)$_{143}$ could be recompounded a second time to maintain the extensibility of 614 ± 155% (Supplementary Fig. S25).

Figure 5 also shows the morphology and interfaces of the pure multiblock, uncompatibilized commercial iPP/LLDPE blend (50:50)

and compatibilized blend with 5 wt.% (PE$_{12}$-$b$-PP$_{10}$)$_{143}$ as investigated by transmission electron microscopy (TEM)[46]. The films were microtomed in liquid nitrogen and stained with RuO$_4$ to enhance contrast between the bright crystalline iPP phase and darker LLDPE phase. While the 50:50 blend of LLDPE/iPP showed phase separation and large droplets of PP within LLDPE (Fig. 5d), the pure multiblock exhibited no discernible microphase separation (Fig. 5c). Worm-shaped lamellae in the multiblock sample can be discerned with average thicknesses of 7 nm and a long period of 15 nm. This corresponds to a $X_c$ of 46% which is in close agreement with the scattering and NMR measurements. Similar homogeneity and high aspect ratio features were reported by Dow in their INFUSE™ multiblock rubbery thermoplastic elastomers of (HDPE-$b$-LLDPE)$_x$[47]. By contrast, INTUNE™ semi-crystalline diblocks phase separate into a co-continuous nanostructured morphology. The shuffled multiblock morphology contains both features of nano-scale homogeneity and a degree of crystallinity >50%, which to the best of our knowledge is a combination of properties not previously reported in polyolefin block copolymers. We recently observed phase separation in LLDPE-$b$-iPP diblock copolymers from living polymerization, and this was only disrupted when the iPP fraction was very small (i.e., 15%)[28]. Since the shuffled multiblocks are 50 wt.% iPP, we conclude that the homogeneous semi-crystallinity is due to the highly segmented nature of the shuffled materials.

Another finding from the TEM is related to the composition of the interface in the blends. Chaffin et al. discovered the interface of Ziegler–Natta iPP and LLDPE consists of amorphous low-molecular-weight chains that are produced during the multi-site polymerization process[48]. Figure 5d captures the same observation in our hands, wherein a dark band appears at the interface of droplets, indicated by the red arrow. When the semi-crystalline multiblock is included in the blend at 5 wt.% loading (Fig. 5e), this amorphous band could not be identified, and the interfaces were more diffuse (see Supplementary Fig. S21 for additional TEM images). As a result, we propose that the semi-crystalline multiblock interphase effectively imbibes these chains into its semi-crystalline morphology as depicted in Fig. 1.

The proposed compatibilization mechanism of these segmented semi-crystalline/semi-crystalline multiblocks combines entanglement and crystallization effects. It has previously been established that the optimal interfacial adhesion between amorphous polymers requires the additive exceeds entanglement molecular weight ($M_e$) five-fold[49]. However, this relationship does not hold in semi-crystalline polymers such as PE ($M_e$ = 1250 g/mol)[49,50] and iPP ($M_e$ = 6300 g/mol)[51], due to the segmental volume that occupies crystalline lamellae, rather than entangling[38]. As such, these shuffled multiblocks form trapped crystallites at the interface, which enhance the yield stresses of the compatibilized blends. The extensibility of the blends is not achieved unless the interfacial chain ties between the bulk polyolefin lamellae, and not simply at the interface, though. Similar effects were articulated by the thread-the-needle mechanism proposed in amorphous phases[16]; and the phenomenon was more generally observed in amorphous systems by Dadmun and coworkers, wherein heptablocks had lower interfacial fracture toughness than equimolecular weight pentablocks[26,27].

## Discussion

The chain-shuffling approach provides access to complementary architectures to living polymerization or chain-shuttling polymerization. While living polymerization benefits precise dispersities and large block sizes, it is operationally difficult to target larger numbers of blocks or lower molar masses. Chain-shuttling addresses both limitations due to stoichiometric low-cost chain transfer moieties (e.g., alkyl–Zn or alkyl–Al), which reversibly transmetalate the growing chain between catalysts with different monomer selectivities. However, no iso-specific catalysts have been shown to be sufficiently selective between ethylene and propene for producing semi-crystalline blocks, hence the presence of amorphous segments in previous systems. The

chain-shuffling strategy described herein separates the polymerization mechanism from the multiblock assembly, which increases versatility and potentially scalability as a non-living polymerization method for polyolefin block copolymers. Despite avoiding stoichiometric living polymerization initiators, the precious metals of the chain-shuffling approach must be recovered to practice this strategy at scale. Therefore, commercial scales will require applying the advances in recyclable metathesis catalysts and progress in the catalytic productivity of telechelic iPP chains.

From the standpoint of compatibilizer design, the results showed that segmented block copolymers require sufficiently long chain-length to entangle into the bulk polyolefin network as tie-chains. Furthermore, we experimentally support the hypothesis that semi-crystalline copolymer interphases imbibe amorphous chains through interfacial crystallization and can be applied towards the enhancement of yield stresses in polyolefin blends, while simultaneously restoring extensibility and toughness. With this foundation of chain-shuffling, previously inaccessible multiblock compositions and architectures from independently synthesized polymers can be envisioned for testing the design limits of plastic blends.

## Methods

### Materials

Toluene was purchased from MilliporeSigma (> 99% pure) and was purified using a Grubb's purification solvent system. Propylene (polymerization grade) and ethylene (polymerization grade) were purchased from Matheson and passed over a stainless-steel pressure tube containing 4 Å molecular sieves and copper deoxygenation catalyst in a 2:1 ratio. Butadiene (> 99%) was purchased from Airgas, PMAO-IP was purchased from Nouryon as a 15% solution in toluene, dimethylsilylbis(1-indenyl)zirconium dichloride was purchased from Strem chemicals, and all other reagents or catalysts were purchased from Sigma Aldrich, which were used as received. INTUNE™ (hard-hard) and DOWLEX™ 2247 G LLDPE (density = 0.917 g/mL; MFI = 2.3 g/10 min) were provided by Dow Chemical. PP4792E1 iPP homopolymer was provided by ExxonMobil (MFI = 2.7 g/10 min).

### Synthesis

*Copolymerization of propylene and butadiene:* In a nitrogen-filled glovebox, a Fisher–Porter bottle equipped with a stir bar was added toluene (100 mL) and PMAO-IP (15% in toluene, 4.16 g, 20 mmol). The vessel was sealed and taken outside the glovebox and pressurized with 1,3-butadiene until 4.0 g had condensed. The vessel was subsequently pressurized with propylene (275 kPa) until the solution was saturated. Separately in the glovebox, a solution of dimethylsilylbis-(1-indenyl)zirconium dichloride (9 mg, 20 μmol) in toluene (3 mL) was prepared and combined with additional PMAO-IP (15% in toluene, 4.16 g, 20 mmol). This solution was loaded into a syringe and sealed with a needle and septum. The prepared catalyst solution was added to the reaction vessel, and the vessel was kept open to a constant 275 kPa pressure of propylene. The reaction was allowed to run for 3 h after which it was quenched with the addition of 5% HCl in methanol, followed by precipitation in excess methanol (ca. 200 mL). The product was filtered and dried overnight under a vacuum oven at 55 °C to yield 6.5–7.1 g polymer over five repeated procedures.

### Hydrogenation of polybutadiene

In a nitrogen-filled glovebox, to a Fisher–Porter bottle equipped with a stir bar was added a prepared toluene solution of polybutadiene (200 mL of 20 g/mL solution, 4 g) and Wilkinson's catalyst (45 mg, 49 μmol) was added along with triphenylphosphine (43 mg, 163 μmol). The reaction vessel was sealed and taken outside the glovebox. The vessel was equilibrated to 75 °C, and then hydrogen gas (380 kPa) was introduced. The reaction was stirred for 5.5 h, cooled to room temperature, and the hydrogen gas was careful vented off with repeated

flushes of nitrogen. The polymer was then precipitated in excess methanol (ca. 200 mL), filtered, and dried overnight in a vacuum oven at 55 °C to yield quantitative isolation of unsaturated PE.

### Ethenolysis of unsaturated polyolefins

To an oven-dried Fisher–Porter bottle was added unsaturated polyethylene (4.0 g) or uPP (6.5 g). The vessel was taken into a nitrogen-filled glovebox, and toluene (150 mL) was added. The reaction vessel was sealed, taken outside the glovebox and heated at 140 °C to first dissolve the unsaturated polyolefin. After the polymer had been dissolved, the temperature of the vessel was lowered to 90 °C. In the glovebox was prepared a solution of the metathesis catalyst tricyclohexylphosphine[3-phenyl-1H-inden-1-ylidene][1,3-bis(2,4,6-trimethylphenyl)-4,5-dihydroimidazol-2-ylidene]ruthenium(II)dichloride (15 mg for unsaturated polyethylene and 20 mg for uPP) in toluene (3 mL), which was then taken up in a sealed syringe. The original reaction vessel was opened to ethylene (70 kPa), and the catalyst solution was injected. The reaction continued for 4 h at 90 °C and constant ethylene pressure, cooled to room temperature, vented of pressure, and poured into excess methanol (200 mL), filtered, and dried overnight in a vacuum oven at 55 °C to yield telechelic polyolefins.

### Chain-shuffling metathesis polymerization

To a flame-dried Schlenk flask equipped with a stir bar was added telechelic di-allyl polyethylene (1.3 g) and telechelic di-allyl polypropylene (1.3 g), and transferred to a nitrogen-filled glovebox. To the vessel was added toluene (1.5 mL) and sealed using a glass top greased with Krytox™ grease and a Teflon™ sealed side-arm valve. The reaction vessel was transferred outside the glovebox and heated to 100 °C with stirring until the reactants dissolved. Inside the glovebox was prepared a separate solution of ruthenium catalyst (Fig. 3) in toluene (catalyst in 1.5 mL of toluene) and loaded into a sealed syringe. This catalyst solution was removed from the glovebox and injected into the reaction mixture under a positive pressure of nitrogen. The reaction was run for 30 min under a flow of nitrogen (35 kPa) fed through the side-arm at 100 °C with a ventilation needle to allow for gas evaporation. After the solvent was completely evaporated, the powdery reaction mixture was redissolved in toluene (5 mL), precipitated using excess methanol (100 mL), filtered, and dried under vacuum overnight in a vacuum oven at 55 °C to yield 2.6 g of multiblock products.

### Characterization

Molecular weights ($M_n$ and $M_w$) and molecular weight distributions ($Đ = M_w/M_n$) were determined by gel permeation chromatography size-exclusion chromatography (GPC/SEC). Analyses were performed using a Tosoh EcoSEC® High Temperature GPC with an RI Detector with two TSKgel GMH$_{HR}$-H HT2 columns (7.8 mm ID × 30 cm, 20 μm) and one 17367-TSKgel Guard Column (7.8 mm ID, 13 μm). Inhibitor-free HPLC grade trichlororbenzene containing 0.01 wt% 3,5-di-*tert*-butyl-4-hydroxytoluene (BHT) was used as the eluent at a flow rate of 1 mL/min at 140 °C. Data were measured relative to polystyrene standards ($Đ < 1.05$) that were adjusted using isotactic polypropylene's Mark–Houwink factors (PS in TCB: $K = 12.1 × 10^{-5}$ dL/g and $α = 0.707$; iPP in TCB: $K = 15.6 × 10^{-5}$ dL/g and $α = 0.760$).

Polymer melting temperature ($T_m$)s were measured by differential scanning calorimetry using DSC-TA Discovery DSC 250. Analyses were performed in aluminum pans under nitrogen, and data were collected from the second heating run at a heating rate of 10 °C/min from 40 to 180 °C and processed with TA TRIOS Software using midpoint half height analysis value.

Materials were processed by a batch microcompounder at 185 °C and compression molded into tensile specimens at 180 °C. The samples were strained at a rate of 100 %/min at 25 °C. Compression molding was carried out using a 4120 hydraulic unit Carver press and

stainless-steel die molds with polyimide protective sheets. Uniaxial tensile elongation was carried out using an Instron series 5566 testing system equipped with a 1 kN load cell and analyzed using BlueHill 3 software. Melt blends were prepared using a vertical conical counter-rotating twin screw batch compounder with a 2.5 mm diameter extrusion die and 5 g capacity mixing chamber. All polymer processing was carried out on as-received materials (i.e., no BHT, other anti-oxidants, or additives were added).

### NMR procedures

Solution $^1$H NMR spectra were recorded using a Varian Inova (400 MHz) spectrometer at 125 °C and were referenced versus residual non-deuterated solvent shifts ($C_2DHCl_4$ δ = 6.0 ppm) ($^1$H). $^{13}$C NMR spectra were recorded on a Varian Inova (750 MHz) spectrometer and were referenced versus solvent shifts ($C_2DHCl_4$ δ = 73.78 ppm) ($^{13}$C).

The $^1$H and $^{13}$C ssNMR experiments were conducted on a BRUKER AVANCE 300 spectrometer equipped with a 4 mm VT CPMAS NMR probe. The carrier frequencies for $^1$H and $^{13}$C were 300.1 MHz and 75.6 MHz, respectively. The $^1$H broad line NMR spectra were recorded using a solid echo sequence with a $^1$H 90° pulse length of 2.2 μs and a delay time of 7 μs. The recycle delay was set to 4 s, with an accumulation number of 64. Each $^1$H NMR experiment lasted about 4 min, with a 5-min interval between experiments. In the $^{13}$C MAS NMR experiment, the MAS frequency was set to 4000 ± 3 Hz. The 90° pulses for $^1$H and $^{13}$C were adjusted to 3.3 μs and 4.5 μs, respectively. The recycle delay and ramp cross-polarization (CP) time were 2 s and 1 ms, respectively. High-power $^1$H TPPM decoupling, with a field strength γB$_1$/2π of 75.8 kHz, was employed during the $^{13}$C acquisition. The chemical shift was referenced to the CH signal of adamantane (29.5 ppm). $^{13}$C direct polarization magic-angle spinning (DPMAS) NMR spectra under high-power $^1$H decoupling were recorded, where the recycle delay was set to a time greater than five times the longest $^{13}$C spin–lattice relaxation time in the laboratory frame ($T_{1C}$). The $^{13}$C spin–lattice relaxation time in the laboratory frame ($T_{1C}$) was measured using the Torchia sequence with a $^{13}$C 90° pulse length of 4.5 μs. The $^1$H spin–lattice relaxation time in the rotating frame ($T_{1ρH}$) was analyzed under a spin-locking field strength of 62.5 kHz.

### Scanning transmission electron microscopy (STEM) procedures

A Tecnai Osiris STEM instrument (FEI, USA) was operated at an acceleration voltage of 200 kV to observe the lamellar structures and the phase-separated morphologies of the samples. The samples were embedded in light-curable acrylic resin (LCR D-800, Toagosei Co., Ltd.) and stained with ruthenium tetroxide ($RuO_4$) for 8 h at 50 °C after creating a flat surface using an ultramicrotome (Leica Ultracut UCT, Wetzlar, Germany). Cross-sections approximately 100 nm thick were then cut by ultramicrotomy with a diamond knife.

## Data availability

The NMR data generated in this study have been deposited in the Figshare database under accession code 10.6084/m9.figshare.30349549. The DSC, GPC, tensile, modeling, and WAXS data generated in this study are provided in the Supplementary Information/Source Data file. All data are available from the corresponding author upon request. Source data are provided with this paper.

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

## Acknowledgements

Financial support is acknowledged from the REMADE Institute and the U.S. Department of Energy's Office of Energy Efficiency and Renewable Energy (EERE) under the Advanced Manufacturing Office Award Number DE-EE0007897, A.B., N.K., and J.E. This research used resources of the Advanced Photon Source, a U.S. Department of Energy (DOE) Office of Science User Facility operated for the DOE Office of Science by Argonne National Laboratory under Contract No. DE-AC02-06CH11357, A.S. and M.D.F. The authors gratefully acknowledge support from the National Science Foundation (NSF) grant NRT-2152210, H.P. and F.K. Financial support was further provided by NSF Division of Materials Research, Polymers, grant 2004393, TM, WGR.

## Author contributions

A.B.: methodology, validation, formal analysis, writing—original draft, data curation, investigation. N.K.: methodology, validation, formal analysis, writing—original draft, visualization, data curation, investigation. W.G.R.: methodology, validation, formal analysis, data curation, investigation, writing—review and editing. H.P.: data curation, methodology, validation. formal analysis, writing—original draft. S.H.: investigation, methodology, validation, formal analysis. A.S.: data curation, investigation, methodology, formal analysis. F.K.: data curation, conceptualization, methodology, writing—review and editing, visualization, resources. M.D.F.: data curation, conceptualization, methodology, investigation, writing—review and editing, visualization, resources. T.M.: data curation, conceptualization, methodology, investigation, writing—review and editing, visualization, resources. J.M.E.: conceptualization, resources, methodology, validation, writing—original draft, writing—review and editing, supervision, project administration, visualization, formal analysis.

## Competing interests

The authors declare no competing interests.
