## [Transparent Peer Review file · Nature Communications]

Segmented Multiblock Polyolefin Compatibilizers from Non-Living Metathesis Chain-Shuffling

Corresponding Author: Professor James Eagan

Version 0:

Reviewer comments:

Reviewer #1

(Remarks to the Author)

The manuscript details the results of a multi-pronged investigation of the design, synthesis, characterization and physical, microscopic, spectroscopic, and analytical interrogation of polydisperse multi-block semicrystalline-semicrystalline PE-b-iPP copolymers as PE/iPP blend compatibilizers. The authors have crafted a very well-written document that is both informative and enjoyable to read.

The novelty of the study revolves around the strategy of independently preparing PE and iPP bearing a degree of internal double bond unsaturation, and employing Ru-catalyzed ethenolysis to obtain lower molar mass telechelic PE and iPP with unsaturated vinyl end groups. Step-growth ADMET polymerization of a 1:1 solution mixture using Ru-catalysts now serves to stitch the PE and iPP chains together to form the desired targeted PE-b_iPP multiblock materials. The authors have conducted an intensive and largely thorough characterization of these PE-b-iPP multiblocks and present data that support their use as PE/iPP blend compatibilizers. An intriguing finding is that for a particular multi-block structure and loading, an increase in the Young's modulus of the compatibilized blend can be achieved. This finding is both novel and important for potentially leading to new designs of polymer additives that can provide desirable changes in physical properties of waste plastics to extend lifetimes for multiple use.

Conceptually, the metathesis strategy was originally demonstrated by the seminal work of Fredrickson and Bazan - which is properly cited. The present work extends this concept with targeted production of the unsaturated iPP through copolymerization of propene with butadiene (which was previously reported by Ishihara and Shiono), followed by Ru catalyzed ethenolysis (which is new). So as a conceptual advance for polymer chemistry, it is hard to justify publication in Nature. The authors further do not discuss or acknowledge the real significant limitations of conducting the overall synthetic strategy on any practical scale. Hence, although the described materials are beautifully constructed, there remains the question of whether they represent any practical advance that could be implemented in real world applications. Perhaps the authors can provide a reasonable response to this criticism.

The real strength of the study lies with the very thorough analysis of the structure / property results that are nicely rationalized at the mechanistic molecular and macromolecular level using both theoretical and experimental considerations. As the authors claim, this study can serve as a foundational contribution for the design of highly effective semicrystalline-based blend compatibilizers for PE and iPP - which will continue to represent a significant Holy Grail with importance to society. For this reason, justification for acceptance can be made - after consideration of a few issues that should be fixed or clarified. These are:

1) The final degree of unsaturation present in the multi-block copolymers was not quantified. Certainly, it can be expected that this unsaturation contributes to oxidative degradation, coloring, and crosslinking - all of which undermines the usefulness as a blend compatibilizer.

2) Related to (1), the authors should carry out the hydrogenation of the multi-block copolymers to provide a fully saturated set of materials. Of course, it will be of interest to compare the properties and compatibilization properties of these saturated multi-blocks to those of the present study. It is possible that the saturated analogs are to be the subject of a different investigation - either way, the authors should acknowledge issues related to having the unsaturation present and how it might impact compatibilization utility.

3) Related to (2), the multi-block materials are not stabilized by any addition of antioxidants. It is conceivable that a degree of cross-linking is introduced in the hot melt press production of samples - and this could conceivably be the origin of the increased Young's modulus that is observed. This is a very important issue and the authors should conduct experiments to either rule it out or confirm its occurrence.

4) The authors did not discuss the possible generation of cyclic oligomers / polymers during the ADMET process. These cyclic polymers would also show only internal unsaturation, but contribute strongly to the observed physical properties - even if present in only small percentage.

5) The authors seemed to have settled on a 5 wt% level of incorporation of the multiblock for compatibilization of a 1:1 blend of PE and iPP. Is this the optimized value? What would the minimum %wt value be for achieving a technologically useful 'compatibilized' blend. The authors do not state what the metrics for compatibilization need to be. Certainly 0.5wt% or 1wt% would be much more attractive. Can the authors take the overall conclusions and factor this into a proposed 'next generation' model? At least to generally suggest which component of blend compatibilization mechanism can benefit from further structure optimization. The authors do state in the last sentence "As a result, the toughness/stiffness tradeoff in compatibilizing mixed plastics can be overcome through appropriate semi-crystalline multiblock compatibilizer design." However, is the present design the limit? or does perhaps a further change in block architecture could be preferred. Since the authors have the two independent sources of telechelic PE and iPP, one can imagine other ways in which these can be stitched together. It would make the article conclusion more compelling for being recognized as a foundational advance - rather than as a practical solution.

Reviewer #2

(Remarks to the Author)

This is an interesting study that seeks to develop novel synthetic techniques to produce polyolefin based multiblock copolymers as well as studies to examine how these multiblock copolymers perform as compatibilizers in polymer blends. I find the "shuffling" polymerization an interesting pathway to forming polyolefin-based blocky copolymers. In fact, the majority of the 'results' discussion is focused on the verification of the success of forming the telechelics and multiblock copolymers. The authors do a very good job of providing supporting evidence that the telechelics are nearly all functionalized at both ends and discussing proper reaction conditions for the successful synthesis of these multiblock copolymers. The authors then transition to examining the effectiveness of these copolymers as compatibilizers for LLDPE/iPP blends, where Figure 4 shows that their presence in the blend significantly strengthens the material. Their inclusion in the blends results in stress-strain curves of the blends that are similar to the pure components, indicating that they are compatibilizing the blends.

My main concern of this manuscript, however, is the presented interpretation of the compatibilization results. It seems to this reviewer that the authors tend to over-interpret the data. For instance, the authors conclude that interfacial crystallization plays an important role in the compatibilization, but I find little direct experimental evidence that there is substantial crystallization at the interface. The authors seem to assume that this interfacial crystallization exists. They also conclude that the compatibilizers are integrated into the homopolymer crystalline network as tie chains, resulting in entanglement and co-crystallization - but again provide no experimental evidence (DSC, WAXS, SAXS?) to back this up. It is this reviewer's opinion that this portion of the manuscript (the interpretation of the compatibilization process) needs significant strengthening.

Along these same lines, I would also argue that the final statement in the abstract "the work provides insights into the prioritization of tie-chains, entanglements, and co-crystallization reinforcement mechanisms in reinforcing post-consumer blends" emerges from the current analysis of the compatibilization results and should be revisited when this analysis is strengthened. Similar statements apply to the last line of the conclusion "As a result, the toughness/stiffness tradeoff in compatibilizing mixed plastics can be overcome through appropriate semi-crystalline multiblock compatibilizer design". This version of the manuscript does not provide sufficient detailed interpretation to elucidate molecular design principles of improved compatibilizers, and therefore this statement should be appropriately modified in the next version.

Additional comments:

- 1.) Page 4, line 64 appears to be a typographical error: "...and/or contain rubbery"
- 2.) The TEM particle results are not very convincing. What is the standard deviation of the reported domain sizes? A qualitative inspection of the TEM results does not show a clear difference in particle size. Is the difference in particle size that is reported statistically relevant? A more detailed presentation of this data should be provided to convince a skeptic that any observed domain size changes are outside the uncertainty of the data analysis.

Reviewer #3

(Remarks to the Author)

In their article, J. Eagan et al. show some fine results and a convincing data set that appear innovative enough to envisage

this work being published in Nature Communications. This study presents a novel method for creating segmented multiblock polyolefin compatibilizers from linear low-density polyethylene (LLDPE) and isotactic polypropylene (iPP) using a non-living metathesis chain-shuffling approach. Key findings are as follows.

1) The method generates semi-crystalline linear multiblock copolymers with tunable block lengths and crystallinity. 2) These copolymers compatibilize and toughen blends of LLDPE and iPP, enhancing mechanical properties like tensile modulus and toughness.

3) The multiblocks reinforce blends through trapped entanglements, tie-chain formation, and interfacial crystallization. 4) These multiblocks thus outperform analogous diblocks, offering superior interfacial mixing and enhanced crystallinity. In the end, the developed methodology is a promising advancement for addressing mixed plastic waste and improving mechanical recycling efforts.

The article is pleasant to read and the results are well presented. The authors examined the properties of the multiblock agents through a multitude of characterizations, including WAXS, SAXS, solution NMR, solid NMR, DSC and mechanical analysis, to account for the compatibilizing effect of mixing LLDPE and iPP. It is however important to point out that the reviewer is not an expert in the field of polyolefin synthesis and the use of derived copolymers as compatibilizing interface agents for homopolymer blends. In fact, for a novice, the reviewer thinks the innovation and the main results of this work were well understood. Obviously, the opinion of more seasoned experts in the field is required to better judge the relevance of this contribution. The following comments can be taken into account for possible improvement of the work, without it being absolutely essential.

- This non-living metathesis chain-shuffling polymerization is efficient in achieving multiblocks. However, it introduces C=C double bonds in the main copolymer chains. There is no comment on that point at all in the paper. This can be a limitation of the method, as these C=C bonds can be oxidized or undergo side reactions upon blending in the micro compounder.

- Following up the previous point, post-chemical modification can be an option to transform these double bonds afterwards.

- Scheme 1 shows the expected structure of both telechelic iPP and LLDPE. As the authors are aware, ADMET polymerization is a step-growth process for which stoichiometry matters a lot. In this case, it is crucial to have telechelics of high -if not perfect- chain-end fidelity. Again, post-chemical modification, for instance, by a thiol-ene reaction followed by thorough characterization would strengthen the assertion of the production of well-functionalized iPP and LLDPE.

- Page 8, about the conditions for the ADMET chain-shuffling polymerization. It is noted that reactions were performed using toluene at 100°C, and that the solvent is then evaporated. Does this mean that the reaction takes place under heterogeneous conditions in bulk? Can this part be clarified by the authors? Can it be envisaged to use another solvent of higher boiling point to favor the ADMET reactions?

- Pages 8-9, Figure 2, Table 1, and elsewhere, about the determination of the MW of the multiblocks. It is important to specify that the MWs are apparent values only, as they are determined by SEC. A short mention about the difficulty to measure absolute MW of polyolefins values can be made. The polymer standards used for calibration must also be mentioned both in the text and in Figure 1 and Table 1.

- The principle of interfacial reinforcement mechanisms could be better illustrated, in Figure 1 or in Figure 5 or both. For a very broad readership like that of Nature Communications, it would be helpful to show a more detailed schematic representation of the trapped entanglements and penetration of the HMW additive into the polyolefin network.

- From the standpoint of chemistry results. In complement to Scheme 1, page 6, it would be worth providing a Figure made of panels showing characteristics of polyolefin and related oligomers, as obtained by SEC, NMR and DSC.

- Page 15. The blends were mixed with a counter-rotating batch 325 micro compounder at 185 °C for 7 minutes. Then, tensile tests were carried out. Did the authors try to reprocess the blends several times under the same compounding conditions. That would be interesting to see whether the reinforcement by the multiblock copolymers can be maintained over multiple processing steps. This remark also echoes to the first point raised about the presence of C=C bonds, i.e. the underlying question is whether the multiblocks are degraded upon heating.

Version 1:

Reviewer comments:

Reviewer #1

(Remarks to the Author)

The revised manuscript incorporates additional results and text that adequately address concerns raised by this reviewer for the original contribution. The authors correctly point out that the aim of the work is not to provide a commercially-viable solution that is ready to be implemented, but rather, to lay an important scientific foundation for new compatibilizer models and mechanisms that can potentially be improved upon through further investigation and optimization. The science being presented is compelling in achieving this goal and the manuscript and results represent a significant contribution to the field of polymer science, technology, and engineering. Recommendation can be provided for acceptance of the revised manuscript and supporting information.

Reviewer #2

(Remarks to the Author)

It appears that the authors have sufficiently addressed the reviewers concerns.

Reviewer #3

(Remarks to the Author)

The authors present a revised manuscript entitled "Segmented Multiblock Polyolefin Compatibilizers from Non-Living Metathesis Chain-Shuffling". I have carefully examined the revised submission along with the detailed responses to the reviewers. The revision represents a significant improvement over the original manuscript. In particular, the authors have clarified the synthetic route and provided convincing evidence for block fidelity and random segmentation. They have also strengthened the mechanistic discussion of compatibilization, highlighting the combined roles of tie-chains, trapped entanglements, and co-crystallization. Claims are supported with a wide range of characterization methods (solution and solid-state NMR, SAXS/WAXS, DSC, TEM, tensile testing). Most of all, the main concern regarding potential crosslinking due to the presence of the double bonds along the compatibilizer backbone has been clearly addressed, demonstrating these effects to be negligible.

The revised manuscript presents a substantial contribution to the field of polymer science and mechanical recycling. It introduces a new synthetic method for compatibilizers that address the challenge of toughening mixed PP/PE blends. I do not see any further major concerns that require additional revision. I therefore support the acceptance of this manuscript in its current form.

Response to Reviewers of “Segmented Multiblock Polyolefin Compatibilizers from Non-Living Metathesis Chain-Shuffling”

We thank the reviewers for their comments and constructive criticisms of our report. We have taken their points into consideration and further developed our understanding of several key points. Most notably, this includes major revisions to: A) the role of unsaturated alkenes in the multiblocks, B) the strength of the evidence of co-crystallization, and C) the limits of compatibilization efficiency with this strategy. We provide explanatory text to these itemized points below. This is in addition to more detailed point-by-point responses to all reviewer comments, which are presented in a table format. We appreciate the opportunity to address these critical scientific aspects and believe it has greatly improved the quality of the manuscript and scientific rigor of our conclusions.

Major Revision A) The role of unsaturated alkenes in the multiblocks

Several reviewers correctly pointed out that polyunsaturated polymers are well known to exhibit degradation through oxidative, chain-scission, and crosslinking reactions. If these occurred in the unsaturated multiblocks described in the manuscript, it would be an alternate explanation for the observed modulus increase and limit the usefulness of these additives in recycling. To address this null hypothesis, sol/gel measurements, stability studies, and post-polymerization modification of the alkenes were carried out to better understand the impact unsaturation has on the system.

First, the high molecular weight multiblock (PP₁₀-*b*₁₃-PE₁₂)₁₄₃ was processed through several conditions aimed at simulating aging. The multiblock was subjected to the same melt-blending procedure used in preparing the PE-PP blends (185 °C, micro-compounder, 7 minutes, N₂), heating in xylenes (130 °C, 96h, air), and prolonged ambient storage in air (20 months, air). Figure S22 (below and added to SI) shows the resulting mass of the soluble and insoluble fractions after continuous Soxhlet extraction in toluene. Notably, no significant (ca. <5%) gelation occurred under any of these conditions, indicating that crosslinking from the alkene moieties is suppressed even in the absence of any added stabilizing chemicals. Additionally, the high temperature SEC analysis (Fig. S22) of the aged materials showed no significant changes in molecular weights (<6%) which could be attributed to alkene reactivity.

Aging conditions	Crude wt.	Wt.% PhMe soluble	M_w (M_w/M_n)
None (as synthesized)	-	-	143,400 (3.7)
Air (20 months)	200 mg	98%	137,800 (3.2)
Xylenes, 130 °C, 96h, air	202 mg	>99%	149,000 (3.9)
Melt, 185 °C, 7m, N ₂	160 mg	96%	151,200 (3.6)

Added Figure S22. High temperature SEC analysis of aged polyunsaturated multiblocks indicating no significant crosslinking, chain-scission, or degradation. Calibrated against PS standards with Mark–Houwink corrections adjusted for the iPP block between 15–23 minutes; a solvent degassing signal is observed at 24 minutes.

Next, we investigated the intentional modification of the internal unsaturation. With a calculated 1.1 mol% olefins in the polymer backbone, these internal alkenes proved recalcitrant to a variety of post-polymerization modification strategies in our experiments. Attempts to hydrogenate with heterogeneous catalysts (Pd/C or Pd/CaCO₃) resulted in <5% conversion of the unsaturated moieties under conditions up to 100 °C and 3.4 MPa of H₂ for 24h, where the materials were fully soluble. Even the highly active Crabtree’s catalyst (i.e., [Ir(cod)(PCy₃)(py)]PF₆) resulted in only modest conversion. Of the conditions tried, only stoichiometric transfer hydrogenation with *p*-TsNHNH₂ (20 mol equiv.) afforded >90% conversion of alkenes to the saturated multiblock polymer (see added Fig S23, below).

This low reactivity is suspected to arise from microstructural and macromolecular steric effects which also contribute to the alkene stability during use (as described above). When this stability is considered in combination with the low concentration of alkenes in the block copolymer, which is notably further diluted in the compatibilized blends, the contributions of unsaturation to coloration, modulus, or degradation were judged to be negligible. This is further supported by the observation that the hydrogenated multiblock exhibits toughened mechanical compatibilization efficacy similar to the unsaturated precursor (Fig. S24).

Added Figure S24. The comparison of $(PE_{12}-b_{13}-iPP_{10})_{143}$ compatibilization as the polyunsaturated sample (A) to the results using the same loading (1 wt.%) of hydrogenated compatibilizer from pTsNHNH₂ (B).

Based on the combined results of these experiments, we can now confidently conclude that the unsaturated structures do not contribute to the reinforcement mechanisms, nor do they detract from the recyclability of the blends due to the extremely low concentration and steric environment between the polyolefin blocks. The results from this major revision are now discussed on Page 20 of the manuscript.

Major Revision B) The evidence of co-crystallization

Crystalline-crystalline block copolymers, particularly polyolefins, are much less studied than their amorphous counterparts. As such the role of co-crystallization in compatibilization is not established and the multiblock materials described are a unique tool for investigating these effects. However, the reviewers fairly criticized the limited evidence for co-crystallization which we have addressed with DSC studies. The revised manuscript now includes the following text on page 19 and a revised Figure 4:

“In order to establish if the PE and PP blocks co-crystallize with the commercial homopolymers, the $(PP_{10}-b_{13}-PE_{12})_{143}$ multiblock copolymer was solvent blended with Ziegler-Natta LLDPE or iPP in a 1:1 ratio and studied by DSC. The commercial

polymers exhibited characteristic melting points (LLDPE $T_m = 107, 125$ °C; iPP $T_m = 168$ °C) and crystallization temperatures (LLDPE $T_c = 102$ °C; iPP $T_c = 117$ °C) and the multiblock compatibilizer exhibited T_m s and T_c s for both phases (LLDPE-block $T_m = 119$ °C, $T_c = 106$ °C; iPP-block $T_m = 131$ °C, $T_c = 99$ °C) – transitions which were assigned based on their telechelic precursors (Fig. 3b). Fig. 4b shows the second heating and first cooling cycles for these materials and their blends. When the multiblock (PP_{10-b}13-PE₁₂)₁₄₃ was blended with the commercial LLDPE, the T_c and T_m of the commercial materials shifted towards the higher values of the multiblock (107 and 121 °C, respectively). This indicates the multiblock induces the PE crystallization process in the commercial LLDPE; meanwhile the iPP phase is unaltered and crystallizes separately. Conversely, when the multiblock was blended with commercial iPP which has higher tacticity, the multiblock iPP T_m of 131 °C and T_c of 99 °C both disappeared. Instead, two melting transitions were observed at 155 and 162 °C and a single T_c of 113 °C was detected indicative of the co-crystallized morphologies, in this case dominated by the Ziegler-Natta iPP material. The PE blocks were unaltered in the PP blend. These DSC results of a model interface support the conclusion that co-crystallization between the multiblock compatibilizer occurs in the compatibilized blends.”

-revised Figure 4 on next page-

Revised Figure 4. (a) Uniaxial tensile tests at 100% min⁻¹ of commercial Ziegler-Natta LLDPE, iPP, blends (50:50), and with the addition of multiblock compatibilizers. (b) DSC of the first cooling and second heating cycles (10 °C min⁻¹) for LLDPE, iPP, multiblock (PE₁₀-b₁₃-PE₁₂)₁₄₃ and their 1:1 blends demonstrating the co-crystallization of the multiblock phases with their respective commercial polyolefins. (c) Transmission electron microscopy of (a) pure multiblock copolymer (PE₉-b₈-iPP₉)₇₇, (d) uncompatibilized blend of LLDPE/iPP (1:1 wt. ratio), and (e) the same blend with 5 wt.% of added (PE₉-b₈-iPP₉)₇₇ multiblock copolymer. Red arrows added to draw attention to the interfaces containing (d) amorphous band of accumulated amorphous materials versus the (e) diffuse interface induced by semi-crystalline multiblock additive.

Major Revision C) The limit of compatibilization efficiency

To test the practicality of these compatibilizers in recycling applications we have expanded the loading and reprocessing conditions. Specifically, after aging the materials for 20 months in an ambient environment to simulate a use-time and then reprocessing the 1:1 LLDPE/iPP blends containing 1 wt% of $(PP_{10}\text{-}b_{13}\text{-}PE_{12})_{143}$ through a second melt-extrusion cycle and tested. The mechanical properties were nearly identical to the original blend in toughness, strain at break, and yield stress (Figure S25 B, below). Additionally, in order to test the limits of compatibilization, the best performing high molecular weight $(PP_{10}\text{-}b_{13}\text{-}PE_{12})_{143}$ was used in 0.5 wt% and 0.2 wt% loadings. These experiments were accompanied by a statistical loss in strain at break and overall toughness, suggesting that these materials require at least 1 wt% to effectively toughen blends. While this is significantly less than 10% commonly needed for diblocks, we note that it is less efficient than ultra-high molecular weight multiblock compatibilizers produced from living polymerization, which were effective even at 0.5 wt%. [ref 13] However, the use of non-living sub-stoichiometric catalysts is critical to economic competitiveness and practical implication.

Added Figure S25. The compatibilization efficiency of $(PE_{12}\text{-}b_{13}\text{-}iPP_{10})_{143}$ in freshly prepared blends (A), after 20 months of aging and subsequent reprocessing at 185 °C for 7 minutes (B), and decreased compatibilizer loading to 0.5 wt% (C) and 0.2 wt% (D) wherein samples are no longer ductile.

Point-by-point reviewer responses:

Reviewer's Comment	Authors' Response
Reviewer 1	
So as a conceptual advance for polymer chemistry, it is hard to justify publication in Nature. The authors further do not discuss or acknowledge the real significant limitations of conducting the overall synthetic strategy on any practical scale. Hence, although the described materials are beautifully constructed, there remains the question of whether they represent any practical advance that could be implemented in real world applications. Perhaps the authors can provide a reasonable response to this criticism.	As polymer chemistry innovates new reactive functionalities for circular materials, we feel it is imperative not to overlook intuitive, yet inaccessible, materials. In this case a study of multiblocks of the two most abundant thermoplastic microstructures, which has yet to be synthesized, much less utilized. In this synthetic pursuit, we advanced catalysis (pushing the limits of both functional iPP and metathesis ADMET) and articulated novel design parameters for co-crystallization at the interface. Given the breadth of newly accessible multiblocks from this introduced non-living method, the mechanistic understanding gained, and potential impact on recycling the most abundant mixed polymer feedstocks, we feel the work is of interest to the broad scientific community. Nonetheless, we wish to acknowledge the limitations and advances needed from the community to make commercial impacts. We have included the additional language in the concluding paragraph: “Despite avoiding stoichiometric living polymerization initiators, the precious metals of the “chain-shuffling” approach must be recovered to practice this strategy at scale. Therefore, commercial scales will require applying the advances in recyclable metathesis catalysts, and progress in the catalytic productivity of telechelic iPP chains.”
1) The final degree of unsaturation present in the multi-block copolymers was not quantified. Certainly, it can be expected that this unsaturation contributes to oxidative degradation, coloring, and crosslinking - all of which undermines the usefulness as a blend compatibilizer.	The additional sol/gel, reprocessing, and hydrogenation studies (Major Revision A) reveal this concern does not manifest experimentally under the conditions studied. We expect higher than expected stability is imparted by the steric environment, crystallinity, and low degree of unsaturated. We have added the quantitative degree of unsaturation from NMR to the main text (below) and SI (Figure S11).

	“The degree of unsaturation in the multiblock was determined by ¹H NMR to be 1.1 mol% of the overall repeat units (Fig S11), consistent with the degree of unsaturation in the telechelic PE and PP precursors (1.2 and 1.6 mol%, respectively).”
2) Related to (1), the authors should carry out the hydrogenation of the multi-block copolymers to provide a fully saturated set of materials. Of course, it will be of interest to compare the properties and compatibilization properties of these saturated multi-blocks to those of the present study. It is possible that the saturated analogs are to be the subject of a different investigation - either way, the authors should acknowledge issues related to having the unsaturation present and how it might impact compatibilization utility.	We are very grateful for this actionable suggestion from the reviewer and have conducted this series of experiments (Major Revision A) to show that hydrogenation did not significantly impact the compatibilization. All samples elongated >300% with addition of 1 wt% of the hydrogenated multiblock.
3) Related to (2), the multi-block materials are not stabilized by any addition of antioxidants. It is conceivable that a degree of cross-linking is introduced in the hot melt press production of samples - and this could conceivably be the origin of the increased Young's modulus that is observed. This is a very important issue and the authors should conduct experiments to either rule it out or confirm its occurrence.	We appreciate this alternative mechanism, which is a reasonable null-hypothesis we did not disprove in the original manuscript. The additional sol-gel and SEC studies demonstrate no crosslinking in the multiblock during melt-processing, prolonged heating, or extended storage (Major Revision A). Combined with the supporting hydrogenated compatibilization results we can now rule out crosslinked additives as an alternative mechanism.
4) The authors did not discuss the possible generation of cyclic oligomers / polymers during the ADMET process. These cyclic polymers would also show only internal unsaturation, but contribute strongly to the observed physical properties - even if present in only small percentage.	We appreciate this suggestion and have added the following text acknowledging our understanding of other architectures forming. “The ADMET “chain-shuffling” reaction is furthermore dependent on the uPP microstructural 1,4-selectivity and end-group fidelity. Currently, our investigations have no evidence in support of, or excluding, the formation of cyclic oligomers from intramolecular cyclization or hyperbranched structures from the <10% of unsaturated units in the combined telechelics that are 1,2-vinyl

	microstructures. Furthermore, the step-growth polymerization of telechelic chains will be terminated by monofunctional chains acting as capping blocks, according to the Carothers equation (Fig. S26).³⁹ Although unfunctionalized chain-ends were below the limit of spectroscopic detection, it may be supposed that a 6-fold reduction in molecular weight during telechelic iPP synthesis affords 17% monofunctional chains. During the shuffling reaction, these capping blocks will limit the maximum DP at 13, which is in close agreement with the experimental SEC measurements. We therefore expect the DP_w is limited by the iPP telechelic fidelity under these conditions.”
5) The authors seemed to have settled on a 5 wt% level of incorporation of the multiblock for compatibilization of a 1:1 blend of PE and iPP. Is this the optimized value? What would the minimum %wt value be for achieving a technologically useful 'compatibilized' blend. The authors do not state what the metrics for compatibilization need to be. Certainly 0.5wt% or 1wt% would be much more attractive. Can the authors take the overall conclusions and factor this into a proposed 'next generation' model? At least to generally suggest which component of blend compatibilization mechanism can benefit from further structure optimization. The authors do state in the last sentence "As a result, the toughness/stiffness tradeoff in compatibilizing mixed plastics can be overcome through appropriate semi-crystalline multiblock compatibilizer design." However, is the present design the limit? or does perhaps a further change in block architecture could be preferred. Since the authors have the two independent sources of telechelic PE and iPP, one can	The reviewer raises another essential point we did not investigate in the original communication. Although, we would clarify that 5 wt% was investigated by TEM in order to visualize the multiblock effects – 1 wt% was also shown to be equally effective as a toughening agent. As the reviewer suggested we have now studied 0.5 wt% and 0.2 wt% which were shown to be less effective at toughening than 1wt% (Major Revision C). We emphasize that compatibilizing at loadings <1 wt% increasingly relies of processing parameters that optimize shear and mixing element design – which the non-reactive compatibilizer field as a whole requires more investigations into. The reviewer also correctly predicts our own thinking on pushing the limits of toughness/stiffness compatibilization. To provide more context of the foundational advance we have simplified the following text to the concluding paragraph: “With this foundation of “chain-shuffling”, previously inaccessible multiblock compositions and architectures from independently synthesized polymers can be envisioned for testing the design limits of plastic blends.”

imagine other ways in which these can be stitched together. It would make the article conclusion more compelling for being recognized as a foundational advance - rather than as a practical solution.

Reviewer 2

My main concern of this manuscript, however, is the presented interpretation of the compatibilization results. It seems to this reviewer that the authors tend to over-interpret the data. For instance, the authors conclude that interfacial crystallization plays an important role in the compatibilization, but I find little direct experimental evidence that there is substantial crystallization at the interface. The authors seem to assume that this interfacial crystallization exists. They also conclude that the compatibilizers are integrated into the homopolymer crystalline network as tie chains, resulting in entanglement and co-crystallization - but again provide no experimental evidence (DSC, WAXS, SAXS?) to back this up. It is this reviewer's opinion that this portion of the manuscript (the interpretation of the compatibilization process) needs significant strengthening.

This was a helpful criticism of the first submission, which we have addressed with the revised DSC studies (**Major Revision B**). We have revised Figure 5/4 and included explanatory text (see major revision B, pages 19-20). We also acknowledge that more detailed investigations of the crystallization kinetics and growth velocity are due in a more specialized report. However, with the added DSC, in combination with the TEM and ssNMR, we are confident in having proven the presence of co-crystallization processes. Lastly, in order to avoid speculative mechanistic discussion, the final paragraph of the results section has been revised:

“The proposed compatibilization mechanism of these segmented semi-crystalline/semi-crystalline multiblocks combines entanglement and crystallization effects. It has previously been established that the optimal interfacial adhesion between amorphous polymers requires the additive exceeds entanglement molecular weight (M_e) five-fold.⁴⁹ However, this relationship does not hold in semi-crystalline polymers such as PE ($M_e = 1,250$ g/mol)⁴⁹ and iPP ($M_e = 6,300$ g/mol),⁵¹ due to the segmental volume that occupies crystalline lamellae, rather than entangling.³⁸ As such, these shuffled multiblocks form trapped crystallites at the interface which enhance the yield stresses of the compatibilized blends. The extensibility of the blends is not achieved unless the interfacial chain ties between the bulk polyolefin lamellae, and not simply at the interface. Similar effects were articulated by the “thread-the-needle” mechanism proposed

	in amorphous phases;¹⁶ and the phenomenon was more generally observed in amorphous systems by Dadmun and coworkers wherein equimolecular weight heptablocks had lower interfacial fracture toughness than pentablocks.^{26,27}
Along these same lines, I would also argue that the final statement in the abstract “the work provides insights into the prioritization of tie-chains, entanglements, and co-crystallization reinforcement mechanisms in reinforcing post-consumer blends” emerges from the current analysis of the compatibilization results and should be revisited when this analysis is strengthened. Similar statements apply to the last line of the conclusion “As a result, the toughness/stiffness tradeoff in compatibilizing mixed plastics can be overcome through appropriate semi-crystalline multiblock compatibilizer design”. This version of the manuscript does not provide sufficient detailed interpretation to elucidate molecular design principles of improved compatibilizers, and therefore this statement should be appropriately modified in the next version.	With the revised DSC studies that establish the co-crystallization between the multiblocks and the bulk phase we have revised these statements to better reflect the results, rather than speculate on the design (Major Revision B). We agree with the reviewer that the phrasing used in the original manuscript should be revised and we have elected to focus on the factual observations by replacing the text in question with: “Furthermore, we experimentally support the hypothesis that semi-crystalline copolymer interphases imbibe amorphous chains through interfacial crystallization and can be applied towards the enhancement of tensile moduli in polyolefin blends, while simultaneously restoring extensibility and toughness.” We also noted several mechanical values in the main text lacked error bars and deviations, which have been added.
2.) The TEM particle results are not very convincing. What is the standard deviation of the reported domain sizes? A qualitative inspection of the TEM results does not show a clear difference in particle size. Is the difference in particle size that is reported statistically relevant? A more detailed presentation of this data should be provided to convince a skeptic that any observed domain size changes are outside the uncertainty of the data analysis.	We thank the reviewer for this point about statistical significance of the morphology data. Indeed, once the deviations are included, the means overlap. For clarification, we do not assert that the compatibilization is due to droplet size reduction (as previously seen in HDPE/iPP blends). Rather the co-crystallization and the trapped tie-chain behavior of the segmented multiblocks are the dominant mechanisms. We made the following changes to address the clarity of our interpretation of the microscopy:  • Standard deviations of particle size added to the SEM figure (now S27)

	 • Simplified discussion around droplet analysis to: “...exhibited slight reduction in average diameters to $1.92 \pm 0.77 \mu\text{m}$ and $1.67 \pm 0.73 \mu\text{m}$, respectively (figure S27). This droplet reduction is much less than in HDPE/iPP blends¹³ and suggests that droplet size reduction is not solely responsible for toughening these blends. “
Reviewer 3	
This non-living metathesis chain-shuffling polymerization is efficient in achieving multiblocks. However, it introduces C=C double bonds in the main copolymer chains. There is no comment on that point at all in the paper. This can be a limitation of the method, as these C=C bonds can be oxidized or undergo side reactions upon blending in the micro compounder.	We thank the reviewer for this critique and have addressed this in Major Revision A
- Following up the previous point, post-chemical modification can be an option to transform these double bonds afterwards.	Also addressed above in Major Revision A. We agree there is potential to utilize the olefins, and we are exploring the alkenes as both reversible welding adhesives and for localizing isotope labeling at block junctions.
- Scheme 1 shows the expected structure of both telechelic iPP and LLDPE. As the authors are aware, ADMET polymerization is a step-growth process for which stoichiometry matters a lot. In this case, it is crucial to have telechelics of high -if not perfect- chain-end fidelity. Again, post-chemical modification, for instance, by a thiol-ene reaction followed by thorough characterization would strengthen the assertion of the production of well-functionalized iPP and LLDPE.	This is an important limitation that the reviewer offers very helpful suggestions on. While several thiol-ene reactions were studied in our laboratory, we observed selective functionalization of vinylidene moieties and modest conversions of allyl and homoallyl end groups by ¹H NMR. This loss of functionality exaggerated the low spectroscopic concentration for detection, well below 1 mol%. In lieu of this, we have more clearly acknowledged the functionality considerations for readers. The telechelic functionality (ideal = 2) for the LLDPE and iPP is now more rigorously discussed: “Furthermore, the step-growth polymerization of telechelic chains will be terminated by monofunctional chains acting as capping blocks, according to the Carothers equation (Fig. S26).³⁹ Although unfunctionalized chain-ends were below the limit of spectroscopic detection, it may be supposed that a 6-fold reduction in molecular

	weight during telechelic iPP synthesis affords 17% monofunctional chains. During the shuffling reaction, these capping blocks will limit the maximum DP_w at 13, which is in close agreement with the experimental SEC measurements. We therefore expect the DP_w of is primarily limited by the iPP telechelic fidelity under these conditions.”
- Page 8, about the conditions for the ADMET chain-shuffling polymerization. It is noted that reactions were performed using toluene at 100°C, and that the solvent is then evaporated. Does this mean that the reaction takes place under heterogeneous conditions in bulk? Can this part be clarified by the authors? Can it be envisaged to use another solvent of higher boiling point to favor the ADMET reactions?	The reviewer’s understanding is correct, and we have incorporated some aspects of this language into the text for readers to more clearly understand the procedures: “This approach maximized intermixing of the PE and PP phases through solvation and the concentration of the end-groups for promoting the polymerization in what is effectively a bulk solid-state polymerization.” Since this systematic report, we have identified 1,1,2-trichloroethylene to be convenient in this reaction without the need to evaporate solvent. We intend to more extensively report these findings separately and feel it may distract from the materials physically characterized in this study (i.e., different telechelic samples were used). Additionally, the recent bans of TCE usage precludes its usefulness in sustainable polymers, so may be irrelevant to the current submission’s aim.
- Pages 8-9, Figure 2, Table 1, and elsewhere, about the determination of the MW of the multiblocks. It is important to specify that the MWs are apparent values only, as they are determined by SEC. A short mention about the difficulty to measure absolute MW of polyolefins values can be made. The polymer standards used for calibration must also be mentioned both in the text and in Figure 1 and Table 1.	We thank the reviewer for the specific actionable correction and have added the following language to the Figures, Table, and SI: Main text: “It is important to note the apparent molecular weights were measured by size-exclusion relative to monodisperse polystyrene standards adjusted by iPP’s Mark–Houwink parameters.” Table 1: “^aDetermined from high temperature size-exclusion chromatography (SEC) in 1,3,5-trichlorobenzene at 140 °C relative to monodisperse PS standards adjusted by Mark–Houwink factors to iPP.”

	SI: Data were measured relative to polystyrene standards ($\bar{M}_w < 1.05$) that were adjusted using isotactic polypropylene's Mark-Houwink factors (PS in TCB: $K = 12.1 \times 10^{-5} \text{ dL/g}$ and $\alpha = 0.707$; iPP in TCB: $K = 15.6 \times 10^{-5} \text{ dL/g}$ and $\alpha = 0.760$).
- The principle of interfacial reinforcement mechanisms could be better illustrated, in Figure 1 or in Figure 5 or both. For a very broad readership like that of Nature Communications, it would be helpful to show a more detailed schematic representation of the trapped entanglements and penetration of the HMW additive into the polyolefin network.	We are thankful for these clarifying suggestions and have modified Figure 1 with this aim. Specifically, we have emphasized the purpose of this study to focus on trapped multiblock entanglements and co-crystallization of the blocks with their respective phases. - From the standpoint of chemistry results. In complement to Scheme 1, page 6, it would be worth providing a Figure made of panels showing characteristics of polyolefin and related oligomers, as obtained by SEC, NMR and DSC.	We thank the reviewer for this suggestion, but respectfully have not implemented a change. We view the SEC data is best presented in the context of ADMET molecular weight optimization and DSC in the context of crystallinity and co-crystallization. We favor the NMR data presented alongside the integrations, degree of unsaturation calculations, and detailed assignments which requires the added space of the SI. Regrettably, we could not devise a clear Scheme 1 that communicated the compiled data.
- Page 15. The blends were mixed with a counter-rotating batch 325 micro compounder at 185 °C for 7 minutes. Then, tensile tests were carried out. Did the authors try to reprocess the blends several times under the same compounding conditions. That would be interesting to see whether the reinforcement by the multiblock copolymers can be maintained over multiple processing steps. This remark also echoes to the first point raised about the presence	The revised mechanical studies have now investigated reprocessing cycles and we can confirm that the toughness and elongations >600% were maintained after 20 months of sample storage as well as after a second shearing by melt-reprocessing (Major Revision C). We note that no additional compatibilizer was added during the recycling process. Additional discussion around the lack of C=C reactivity has also been added, as detailed above (Major Revision A).

of C=C bonds, i.e. the underlying question is whether the multiblocks are degraded upon heating.	
--	--

Additional edits include:

- The addition of Walter G. Romano as an author who contributed to the co-crystallization studies, multiblock functionalization, and additional mechanical testing.
- The renumbering of references to accommodate newly added reference 39.
- Renumbering and expanding the supplementary table of contents.